# AFEC: Active Forgetting of Negative Transfer in Continual Learning

**Liyuan Wang**[1,2,3]     **Mingtian Zhang**[4]     **Zhongfan Jia**[5]     **Qian Li**[1,2]

**Kaisheng Ma**[5]     **Chenglong Bao**[6]     **Jun Zhu**[3*]     **Yi Zhong**[1,2*]

[1]School of Life Sciences, IDG/McGovern Institute for Brain Research, Tsinghua University.
[2]Tsinghua-Peking Center for Life Sciences. [3]Dept. of Comp. Sci. & Tech., Institute for AI,
BNRist Center, THBI Lab, Tsinghua University. [4]AI Center, University College London.
[5]IIIS, Tsinghua University. [6]Yau Mathematical Sciences Center, Tsinghua University.
`{wly19,jzf20}@mails.tsinghua.edu.cn, mingtian.zhang.17@ucl.ac.uk`
`{liqian8,kaisheng,clbao,dcszj,zhongyithu}@tsinghua.edu.cn`

## Abstract

Continual learning aims to learn a sequence of tasks from dynamic data distributions. Without accessing to the old training samples, knowledge transfer from the old tasks to each new task is difficult to determine, which might be either positive or negative. If the old knowledge interferes with the learning of a new task, i.e., the forward knowledge transfer is negative, then precisely remembering the old tasks will further aggravate the interference, thus decreasing the performance of continual learning. By contrast, biological neural networks can actively forget the old knowledge that conflicts with the learning of a new experience, through regulating the learning-triggered synaptic expansion and synaptic convergence. Inspired by the biological active forgetting, we propose to actively forget the old knowledge that limits the learning of new tasks to benefit continual learning. Under the framework of Bayesian continual learning, we develop a novel approach named Active Forgetting with synaptic Expansion-Convergence (AFEC). Our method dynamically expands parameters to learn each new task and then selectively combines them, which is formally consistent with the underlying mechanism of biological active forgetting. We extensively evaluate AFEC on a variety of continual learning benchmarks, including CIFAR-10 regression tasks, visual classification tasks and Atari reinforcement tasks, where AFEC effectively improves the learning of new tasks and achieves the state-of-the-art performance in a plug-and-play way.

## 1 Introduction

The ability to continually learn numerous tasks from dynamic data distributions is critical for deep neural networks, which needs to remember the old tasks by avoiding catastrophic forgetting [18] while effectively learn each new task by improving forward knowledge transfer [17]. Due to the dynamic data distributions, forward knowledge transfer might be either positive or negative, and is difficult to determine without accessing to the old training samples. If the forward knowledge transfer is *negative*, i.e., learning a new task from the old knowledge is worse than learning the new task on a randomly-initialized network [36, 17], then precisely remembering the old tasks will severely interfere with the learning of the new task, thus decreasing the performance of continual learning.

---

*Corresponding author: J. Zhu and Y. Zhong.

35th Conference on Neural Information Processing Systems (NeurIPS 2021).

By contrast, biological neural networks can effectively learn a new experience on the basis of remembering the old experiences, even if they conflict with each other [18, 5]. This advantage, called *memory flexibility*, is achieved by *active forgetting* of the old knowledge that interferes with the learning of a new experience [29, 5]. The latest data suggested that the underlying mechanism of biological active forgetting is to regulate the learning-triggered synaptic expansion and synaptic convergence (Fig. 1, see Appendix A for neuroscience background and our biological data). Specifically, the biological synapses expand additional functional connections to learn a new experience together with the previously-learned functional connections (synaptic expansion). Then, all the functional connections are pruned to the amount before learning (synaptic convergence).

Inspired by the biological active forgetting, we propose to actively forget the old knowledge that interferes with the learning of new tasks without significantly increasing catastrophic forgetting, so as to benefit continual learning. Specifically, we adopt Bayesian continual learning and actively forget the posterior distribution that absorbs all the information of the old tasks with a forgetting

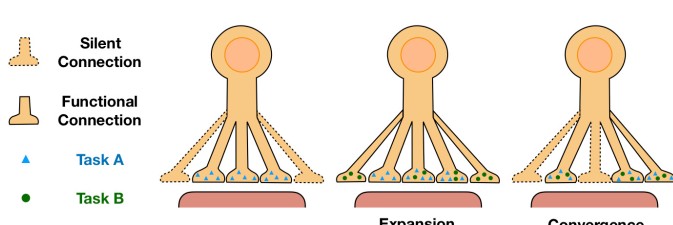

Figure 1: The biological active forgetting is achieved by regulating the learning-triggered synaptic expansion-convergence.

factor to better learn each new task. Then, we derive a novel method named Active Forgetting with synaptic Expansion-Convergence (AFEC), which is formally consistent with the underlying mechanism of biological active forgetting at synaptic structures. Beyond regular weight regularization approaches [12, 1, 35, 2], which selectively penalize changes of the important parameters for the old tasks, AFEC dynamically expands parameters only for each new task to avoid potential negative transfer from the main network, while the forgetting factor regulates a penalty to selectively merge the main network parameters with the expanded parameters, so as to learn a better overall representation of both the old tasks and the new task.

We extensively evaluate AFEC on continual learning of CIFAR-10 regression tasks, a variety of visual classification tasks, and Atari reinforcement tasks [10], where AFEC achieves the state-of-the-art (SOTA) performance. We empirically validate that the performance improvement results from effectively improving the learning of new tasks without increasing catastrophic forgetting. Further, AFEC can be a *plug-and-play* method that significantly boosts the performance of representative continual learning strategies, such as weight regularization [12, 1, 35, 2] and memory replay [21, 9, 6].

Our contributions include: (1) We draw inspirations from the biological active forgetting and propose a novel approach to actively forget the old knowledge that interferes with the learning of new tasks for continual learning; (2) Extensive evaluation on a variety of continual learning benchmarks shows that our method effectively improves the learning of new tasks and achieves the SOTA performance in a plug-and-play way; and (3) To the best of our knowledge, we are the first to model the biological active forgetting and its underlying mechanism at synaptic structures, which suggests a potential theoretical explanation of how the underlying mechanism of biological active forgetting achieves its function of forgetting the past and continually learning conflicting experiences [29, 5].

## 2   Related Work

Continual learning needs to minimize catastrophic forgetting and maximize forward knowledge transfer. Existing work in continual learning mainly focuses on mitigating catastrophic forgetting. Representative approaches include: weight regularization [12, 1, 35, 2], which selectively penalizes changes of the previously-learned parameters; parameter isolation [24, 10], which allocates a dedicated parameter subspace for each task; and memory replay [21, 28, 9], which approximates and recovers the old data distributions through storing old training data, their embedding or learning a generative model. In particular, Adaptive Group Sparsity based Continual Learning (AGS-CL) [10] proposed to regularize the group sparsity with separation of the important nodes for the old tasks to prevent catastrophic forgetting, which takes advantages of weight regularization and parameter isolation, and achieved the SOTA performance on various continual learning benchmarks.

Several studies suggested that forward knowledge transfer is critical for continual learning [17, 4], which might be either positive or negative due to the dynamic data distributions. Although it is highly nontrivial to mitigate potential negative transfer while overcoming catastrophic forgetting, the efforts that specifically consider this challenging issue are limited. [3] developed a method to mitigate negative transfer when fine-tuning tasks on a pretrained network. For the scenario where the old tasks can be learned again, [26] learned an additional active column to better exploit potential positive transfer. [22] tried to maximize transfer and minimize interference from a memory buffer containing a few old training data. Similarly, [6, 16, 33] attempted to more effectively balance stability and plasticity with the memory buffer in class incremental learning, while [32] stored and updated the old features. By contrast, since pretraining or old training data might not be available in continual learning, we mainly focus on a more restrict yet realistic setting that a neural network incrementally learns a sequence of tasks *from scratch*, *without* storing old training data. Further, we extend our method to the scenarios where pretraining or memory buffer can be used, as well as the scenarios other than classification tasks, such as regression tasks and reinforcement tasks.

## 3 Method

In this section, we first describe the framework of Bayesian continual learning [12, 20]. Under such framework, we propose an active forgetting strategy, which is formally consistent with the underling mechanism of biological active forgetting at synaptic structures.

### 3.1 Basics of Bayesian Continual Learning

Continual learning needs to remember the old tasks and learn each new task effectively. Let's consider a simple case that a neural network with parameter $\theta$ continually learns two independent tasks, task $A$ and task $B$, from their training datasets $D_A^{train}$ and $D_B^{train}$ [12]. The training dataset of each task is only available when learning the task.

**Bayesian Learning:** After learning $D_A^{train}$, the posterior distribution

$$p(\theta|D_A^{train}) = \frac{p(D_A^{train}|\theta)p(\theta)}{p(D_A^{train})}$$

incorporates the knowledge of task $A$. Then, we can get the predictive distribution for the test data of task $A$:

$$p(D_A^{test}|D_A^{train}) = \int p(D_A^{test}|\theta)p(\theta|D_A^{train})d\theta.$$

As the posterior $p(\theta|D_A^{train})$ is generally intractable (except very special cases), we must resort to approximation methods, such as the Laplace approximation [12] or other approaches of approximate inference [20]. Let's take Laplace approximation as an example. If $p(\theta|D_A^{train})$ is smooth and majorly peaked around the mode $\theta_A^* = \arg\max_\theta \log p(\theta|D_A^{train})$, we can approximate it with a Gaussian distribution whose mean is $\theta_A^*$ and covariance is the inverse Hessian of the negative log posterior (detailed in Appendix B.1).

**Bayesian Continual Learning:** Next, we want to incorporate the new task into the posterior, which uses the posterior $p(\theta|D_A^{train})$ as the prior of the next task [12]:

$$p(\theta|D_A^{train}, D_B^{train}) = \frac{p(D_B^{train}|\theta)\,p(\theta|D_A^{train})}{p(D_B^{train})}. \tag{1}$$

Then we can test the performance of continual learning by evaluating

$$p(D_A^{test}, D_B^{test}|D_A^{train}, D_B^{train}) = \int p(D_A^{test}, D_B^{test}|\theta)p(\theta|D_A^{train}, D_B^{train})d\theta. \tag{2}$$

Similarly, $p(\theta|D_A^{train}, D_B^{train})$ can be approximated by a Gaussian using Laplace approximation whose mean is the mode of the posterior:

$$\theta_{A,B}^* = \arg\max_\theta \log p(\theta|D_A^{train}, D_B^{train}) \tag{3}$$

$$= \arg\max_\theta \log p(D_B^{train}|\theta) + \log p(\theta|D_A^{train}) - \underbrace{\log p(D_B^{train})}_{const.}. \tag{4}$$

This MAP estimation is also known as the Elastic Weight Consolidation (EWC) [12]:

$$L_{\text{EWC}}(\theta) = L_{\text{B}}(\theta) + \frac{\lambda}{2} \sum_i F_{A,i}(\theta_i - \theta_{A,i}^*)^2, \tag{5}$$

where $L_{\text{B}}(\theta)$ is the loss for task $B$ and $i$ is the label of each parameter. $F_A$ is the Fisher Information matrix (FIM) of $\theta_A^*$ on $D_A^{train}$ (the computation is detailed in Appendix B.1), which indicates the "importance" of parameter $i$ for task $A$. The hyperparameter $\lambda$ explicitly controls the penalty that selectively merges each $\theta_i$ to $\theta_{A,i}^*$ to alleviate catastrophic forgetting.

## 3.2 Active Forgetting with Synaptic Expansion-Convergence

However, if precisely remembering task $A$ interferes with the learning of task $B$, e.g., task $A$ and task $B$ are too different, it might be useful to *actively forget* the original data, similar to the biological strategy of active forgetting. Based on this inspiration, we introduce a forgetting factor $\beta$ and replace $p(\theta|D_A^{train})$ that absorbs all the information of $D_A^{train}$ with a weighted product distribution [8, 19]:

$$p_m(\theta|D_A^{train}, \beta) = \frac{p(\theta|D_A^{train})^{(1-\beta)} p(\theta)^\beta}{Z}, \tag{6}$$

where we use $m$ to denote that we are 'mixing' $p(\theta|D_A^{train})$ and $p(\theta)$ to produce the new distribution $p_m$. $Z$ is the normalizer that depends on $\beta$, which keeps $p_m(\theta|D_A^{train}, \beta)$ following a Gaussian distribution if $p(\theta|D_A^{train})$ and $p(\theta)$ are both Gaussian (detailed in Appendix B.2). When $\beta \to 0$, $p_m$ will be dominated by $p(\theta|D_A^{train})$ and remember all the information about task $A$. When $\beta \to 1$, $p_m$ will actively forget all the information about task $A$. Modified from Eqn. (2), our target becomes:

$$p(D_A^{test}, D_B^{test}|D_A^{train}, D_B^{train}, \beta) = \int p(D_A^{test}, D_B^{test}|\theta) p(\theta|D_A^{train}, D_B^{train}, \beta) d\theta. \tag{7}$$

We first need to determine $\beta$, which decides how much information from task $A$ is forgotten to maximize the probability of learning task $B$ well. A good $\beta$ should be as follows:

$$\beta^* = \arg\max_\beta p(D_B^{train}|D_A^{train}, \beta) = \arg\max_\beta \int p(D_B^{train}|\theta) p_m(\theta|D_A^{train}, \beta) d\theta. \tag{8}$$

Since the integral is difficult to solve, we can make a grid search to determine $\beta$, which should be between 0 and 1. Next, $p(\theta|D_A^{train}, D_B^{train}, \beta)$ can also be approximated by a Gaussian using Laplace approximation (the proof is detailed in Appendix B.3), and the MAP estimation is

$$\begin{aligned}
\theta_{A,B}^* &= \arg\max_\theta \log p(\theta|D_A^{train}, D_B^{train}, \beta) \\
&= \arg\max_\theta (1 - \beta)(\log p(D_B^{train}|\theta) + \log p(\theta|D_A^{train})) + \beta \log p(\theta|D_B^{train}) + const..
\end{aligned} \tag{9}$$

Then we obtain the loss function of Active Forgetting with synaptic Expansion-Convergence (AFEC):

$$L_{\text{AFEC}}(\theta) = L_{\text{B}}(\theta) + \frac{\lambda}{2} \sum_i F_{A,i}(\theta_i - \theta_{A,i}^*)^2 + \frac{\lambda_e}{2} \sum_i F_{e,i}(\theta_i - \theta_{e,i}^*)^2. \tag{10}$$

$\theta_e^*$ are the optimal parameters for the new task and $F_e$ is the FIM of $\theta_e^*$ (the computation is detailed in Appendix B.1). As shown in Fig. 2, we first learn a set of expanded parameters $\theta_e$ with $L_{\text{B}}(\theta_e)$ to obtain $\theta_e^*$ and $F_e$. Then we can optimize Eqn. (10), where two weight-merging regularizers selectively merge $\theta_i$ with $\theta_{A,i}^*$ for the old tasks and $\theta_{e,i}^*$ for the new task. The forgetting factor $\beta$ is integrated into a hyperparameter $\lambda_e \propto \beta/(1-\beta)$ to control the penalty that promotes active forgetting. Therefore, derived from active forgetting of the original posterior in Eqn. (6), we obtain an algorithm that dynamically expands parameters to learn a new task and then selectively converges the expanded parameters to the main network. Intriguingly, this algorithm is formally consistent with the underlying mechanism of biological active forgetting (the neuroscience evidence is detailed in Appendix A), which also expands additional functional connections for a new experience (synaptic expansion) and then prunes them to the amount before learning (synaptic convergence).

As the proposed active forgetting is integrated into the third term, our method can be used in a *plug-and-play* way to improve continual learning (detailed in Appendix E, F). Here we use Laplace approximation to approximate the intractable posteriors, which can be other strategies of approximate inference [20] in further work. Note that $\theta_e^*$ and $F_e$ are *not* stored in continual learning, and the architecture of the main network is *fixed*. Thus, AFEC does not cause additional storage cost compared with regular weight regularization approaches such as [12, 35, 1, 2]. Further, it is straightforward to extend our method to continual learning of more than two tasks. We discuss it in Appendix B.4 with a pseudocode.

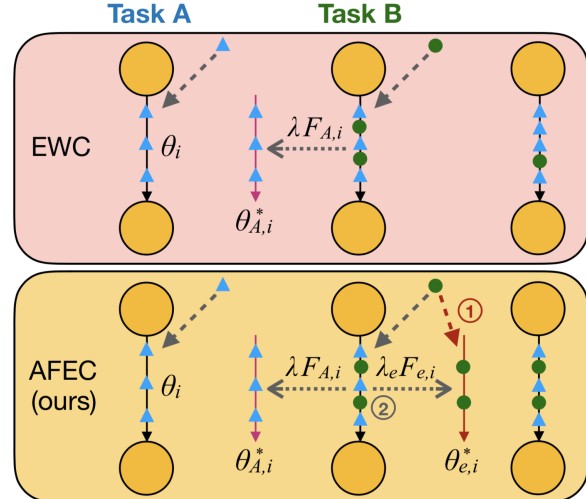

Figure 2: Conceptual comparison of EWC and AFEC (ours). ① *Synaptic Expansion*: Learn the expanded parameters $\theta_e$ with $L_B(\theta_e)$ to obtain $\theta_e^*$ and $F_e$. ② *Synaptic Convergence*: Learn the main network parameters $\theta$ with Eqn. (10) for selective weight-merging.

Now we conceptually analyze how AFEC mitigates potential negative transfer in continual learning (see Fig. 2). When learning task $B$ on the basis of task $A$, regular weight regularization approaches [12, 35, 1, 2] selectively penalize changes of the old parameters learned for task $A$, which will severely interfere with the learning of task $B$ if they conflict with each other. In contrast, AFEC learns a set of expanded parameters only for task $B$ to avoid potential negative transfer from task $A$. Then, the main network parameters selectively merge with both the old parameters and the expanded parameters, depending on their contributions to the overall representations of task $A$ and task $B$.

## 4  Experiment

In this section, we evaluate AFEC on a variety of continual learning benchmarks, including: CIFAR-10 regression tasks, which is a toy experiment to validate our idea about negative transfer in continual learning; visual classification tasks, where the forward knowledge transfer might be either positive or negative; and Atari reinforcement tasks, where the forward knowledge transfer is severely negative. All the experiments are averaged by 5 runs with different random seeds and task orders.

### 4.1  CIFAR-10 Regression Tasks

First, we propose CIFAR-10 regression tasks to explicitly show how negative transfer affects continual learning, and how AFEC effectively addresses this challenging issue. CIFAR-10 dataset [13] contains 50,000 training samples and 10,000 testing samples of 10-class colored images of size $32 \times 32$. The regression task is to evenly map the ten classes around the origin of the two-dimensional coordinates and train the neural network to predict the angle of the origin to each class (see Fig. 3). We change the relative position of the ten classes to construct different regression tasks with mutual negative transfer,

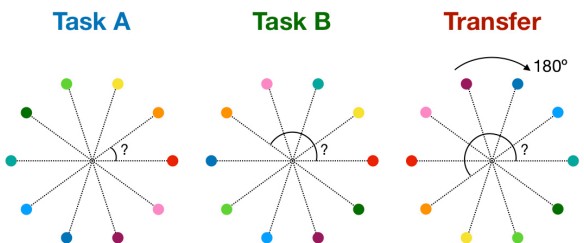

Figure 3: CIFAR-10 regression tasks. Each circle represents the position of a class. Task $A$ and Task $B$ use different relative positions. "Transfer" applies the same relative position as Task $A$, but rotates by several phases.

in which remembering the old knowledge will severely interfere with the learning of a new task.

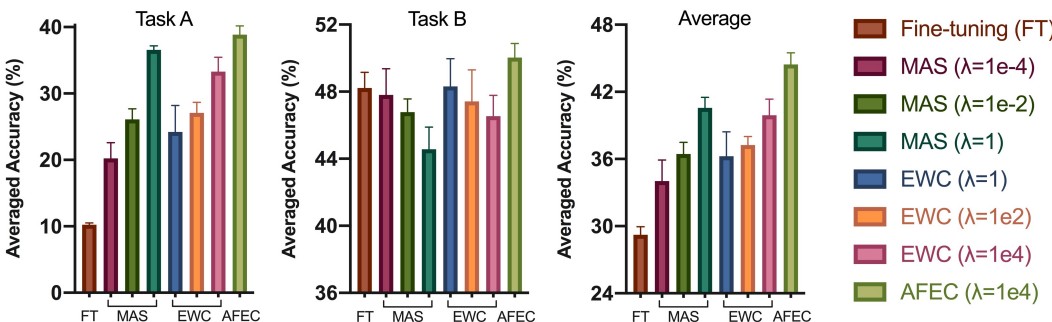

Figure 4: Continual learning of two CIFAR-10 regression tasks with a two-layer LeNet architecture. Larger strength of weight regularization can better remember the old task but limits the learning of the new task. AFEC can more effectively learn a new task while remembering the old task.

As shown in Fig. 4 for continual learning of two different regression tasks, regular weight regularization approaches, such as MAS [1] and EWC [12], can effectively remember the old tasks, but limits the learning of new tasks. In particular, larger strength of the weight regularization results in better performance of the

Table 1: Continual learning of CIFAR-10 regression tasks with various architectures. We present the averaged accuracy (%) of five runs for two-task and ten-task, and five runs of five rotations for transfer experiment.

|  | Methods | LeNet [15] | VGG11 [30] | VGG11BN [30] | ResNet10 [7] |
|---|---|---|---|---|---|
| Two-Task | Fine-tuning | 29.23 $_{\pm 0.72}$ | 46.37 $_{\pm 0.11}$ | 46.54 $_{\pm 0.29}$ | 60.67 $_{\pm 1.52}$ |
|  | EWC [12] | 39.91 $_{\pm 1.44}$ | 73.55 $_{\pm 1.26}$ | 82.00 $_{\pm 0.32}$ | 71.94 $_{\pm 1.61}$ |
|  | AFEC (ours) | **44.45** $_{\pm 1.03}$ | **77.76** $_{\pm 0.09}$ | **86.07** $_{\pm 0.24}$ | **75.67** $_{\pm 1.19}$ |
| Ten-Task | Fine-tuning | 46.57 $_{\pm 0.68}$ | 18.03 $_{\pm 0.03}$ | 18.08 $_{\pm 0.04}$ | 54.97 $_{\pm 1.33}$ |
|  | EWC [12] | 49.95 $_{\pm 1.81}$ | 79.39 $_{\pm 1.12}$ | 85.98 $_{\pm 0.07}$ | 82.91 $_{\pm 0.22}$ |
|  | AFEC (ours) | **53.50** $_{\pm 1.70}$ | **82.50** $_{\pm 0.47}$ | **88.31** $_{\pm 0.11}$ | **85.33** $_{\pm 0.31}$ |
| Transfer | Fine-tuning | 38.93 $_{\pm 0.80}$ | 80.37 $_{\pm 0.40}$ | 84.30 $_{\pm 0.10}$ | 85.69 $_{\pm 0.93}$ |
|  | EWC [12] | 35.87 $_{\pm 0.87}$ | 76.66 $_{\pm 0.44}$ | 82.25 $_{\pm 0.11}$ | 84.96 $_{\pm 0.91}$ |
|  | AFEC (ours) | **40.90** $_{\pm 1.35}$ | **83.81** $_{\pm 0.42}$ | **86.30** $_{\pm 0.17}$ | **87.80** $_{\pm 0.66}$ |

first task but worse performance of the second task. In contrast, AFEC improves the learning of new tasks on the basis of remembering the old tasks, so as to achieve better averaged accuracy. Note that EWC is equal to the ablation of active forgetting in AFEC, i.e., $\beta = 0$, so the performance improvement of AFEC on EWC validates the effectiveness of our proposal. We further demonstrate the efficacy of AFEC on a variety of architectures and a larger amount of tasks (see Table 1).

In addition, we evaluate the ability of transfer learning after continual learning of two different regression tasks. We fix the feature extractor of the neural network and only fine-tune a linear classifier to predict a new task that is similar to the first task. Specifically, the similar task applies the same relative position as the first task, but rotates by $60°$, $120°$, $180°$, $240°$ or $300°$. Therefore, if the neural network effectively remembers and transfers the relative position learned in the first task, it will be able to learn the similar task well. As shown in Table 1, AFEC can more effectively learn the similar task, while EWC is even worse than sequentially fine-tuning without weight regularization.

## 4.2 Visual Classification Tasks

**Dataset:** We evaluate continual learning on a variety of benchmark datasets for visual classification, including CIFAR-100, CUB-200-2011 and ImageNet-100. CIFAR-100 [13] contains 100-class colored images of the size $32 \times 32$, where each class includes 500 training samples and 100 testing samples. CUB-200-2011 [31] is a large-scale dataset including 200 classes and 11,788 colored images of birds, split as 30 images per class for training while the rest for testing. ImageNet-100 [9] is a subset of iILSVRC-2012 [23], consisting of randomly selected 100 classes of images and 1300 samples per class. We follow the regular preprocessing pipeline of CUB-200-2011 and ImageNet-100 as [10], which randomly resizes and crops the images to the size of $224 \times 224$ before experiment.

**Benchmark:** We consider five representative benchmarks of visual classification tasks to evaluate continual learning in different aspects. The first three are on CIFAR-100, with forward knowledge transfer from more negative to more positive (detailed in Fig. 5), while the second two are on large-scale images. (1) CIFAR-100-SC [34]: CIFAR-100 can be split as 20 superclasses (SC) with 5 classes per superclass dependent on semantic similarity, where each superclass is a classification task. Since the superclasses are semantically different, forward knowledge transfer in such a task sequence is

Table 2: Averaged accuracy (%) of all the tasks learned so far in continual learning of visual classification tasks, averaged by 5 different random seeds (see Appendix C for error bar). *AFEC is our method described in Sec. 3.2, while *w/* AFEC is the adaptation of our method to representative weight regularization methods (detailed in Appendix E).

| Methods | CIFAR-100-SC | | CIFAR-100 | | CIFAR-10/100 | | CUB-200 w/ PT | | CUB-200 w/o PT | | ImageNet-100 | |
| | $A_{10}$ | $A_{20}$ | $A_{10}$ | $A_{20}$ | $A_2$ | $A_{2+20}$ | $A_5$ | $A_{10}$ | $A_5$ | $A_{10}$ | $A_5$ | $A_{10}$ |
|---|---|---|---|---|---|---|---|---|---|---|---|---|
| Fine-tuning | 32.58 | 28.40 | 40.92 | 33.53 | 78.96 | 37.81 | 78.75 | 78.13 | 31.91 | 39.82 | 50.56 | 44.80 |
| P&C [26] | 53.48 | 52.88 | 70.10 | 70.21 | 86.72 | 78.29 | 81.42 | 81.74 | 33.88 | 42.79 | 76.44 | 74.38 |
| AGS-CL [10] | 55.19 | 53.19 | 71.24 | 69.99 | 86.27 | 80.42 | 82.30 | 81.84 | 32.69 | 40.73 | 51.48 | 47.20 |
| EWC [12] | 52.25 | 51.74 | 68.72 | 69.18 | 85.07 | 77.75 | 81.37 | 80.92 | 32.90 | 42.29 | 76.12 | 73.82 |
| *AFEC (ours) | **56.28** | **55.24** | **72.36** | **72.29** | **86.87** | **81.25** | **83.65** | **82.04** | **34.36** | 43.05 | **77.64** | 75.46 |
| MAS [1] | 52.76 | 52.18 | 67.60 | 69.41 | 84.97 | 77.39 | 79.98 | 79.67 | 31.68 | 42.56 | 75.48 | 74.72 |
| *w/* AFEC (ours) | 55.26 | 54.89 | 69.57 | 71.20 | 86.21 | 80.01 | 82.77 | 81.31 | 34.08 | 42.93 | 75.64 | **75.66** |
| SI [35] | 52.20 | 51.97 | 68.72 | 69.21 | 85.00 | 76.69 | 80.14 | 80.21 | 33.08 | 42.03 | 73.52 | 72.97 |
| *w/* AFEC (ours) | 55.25 | 53.90 | 69.34 | 70.13 | 85.71 | 78.49 | 83.06 | 81.88 | 34.04 | **43.20** | 75.72 | 74.14 |
| RWALK [2] | 50.51 | 49.62 | 66.02 | 66.90 | 85.59 | 73.64 | 80.81 | 80.58 | 32.56 | 41.94 | 73.24 | 73.22 |
| *w/* AFEC (ours) | 52.62 | 51.76 | 68.50 | 69.12 | 86.12 | 77.16 | 83.24 | 81.95 | 33.35 | 42.95 | 74.64 | 73.86 |

relatively more negative. (2) CIFAR-100 [21]: The 100 classes in CIFAR-100 are randomly split as 20 classification tasks with 5 classes per task. (3) CIFAR-10/100 [10]: The 10-class CIFAR-10 are randomly split as 2 classification tasks with 5 classes per task, followed by 20 tasks with 5 classes per task randomly split from CIFAR-100. This benchmark is adapted from [10] to keep the number of classes per task the same as benchmark (1, 2), where the large amounts of training data in the first two CIFAR-10 tasks bring a relatively more positive transfer. (4) CUB-200 [10]: The 200 classes in CUB-200-2011 are randomly split as 10 classification tasks with 20 classes per task. (5) ImageNet-100 [21]: The 100 classes in ImageNet-100 are randomly split as 10 classification tasks with 10 classes per task.

**Architecture:** We follow [10] to use a CNN architecture with 6 convolution layers and 2 fully connected layers for benchmark (1, 2, 3), and AlexNet [14] for benchmark (4, 5). Since continual learning needs to quickly learn a usable model from incrementally collected data, we mainly consider learning the network from scratch. Following [10], we also try AlexNet with ImageNet pretraining for CUB-200.

**Baseline:** First, we consider a restrict yet realistic setting of continual learning *without* access to the old training data, and perform multi-head evaluation [2]. Since AFEC is a weight regularization approach, we mainly compare with representative approaches that follow a similar idea, such as EWC [12], MAS [1], SI [35] and RWALK [2]. We also compare with AGS-CL [10], the SOTA method that takes advantage of weight regularization and parameter isolation, and P&C [26], which learns an additional active column on the basis of EWC to improve forward knowledge transfer. We reproduce the results of all the baselines from the officially released code of [10], where we do an extensive hyperparameter search and report the best performance for fair comparison (detailed in Appendix C). Then, we relax the restriction of using old training data and plug AFEC in representative memory replay approaches, where we perform single-head evaluation [2] (detailed in Appendix F).

**Averaged Accuracy:** In Table 2, we summarize the averaged accuracy of all the tasks learned so far during continual learning of visual classification tasks. AFEC achieves the best performance on all the continual learning benchmarks and is much better than EWC [12], i.e., the ablation of active forgetting in AFEC. In particular, AGS-CL [10] is the SOTA method on relatively small-scale images and on CUB-200 with ImageNet pretraining (CUB-200 w/ PT). While, AFEC achieves a better performance than AGS-CL on small-scale images from scratch and CUB-200 w/ PT, and substantially outperforms AGS-CL on the two benchmarks of large-scale images from scratch. Further, since regular weight regularization approaches are generally in a re-weighted weight decay form, AFEC can be easily adapted to such approaches (the adaptation is detailed in Appendix E) and effectively boost their performance on the benchmarks above.

**Knowledge Transfer:** Next, we evaluate knowledge transfer in the three continual learning benchmarks developed on CIFAR-100 in Fig. 5. We first present the accuracy of learning each new task

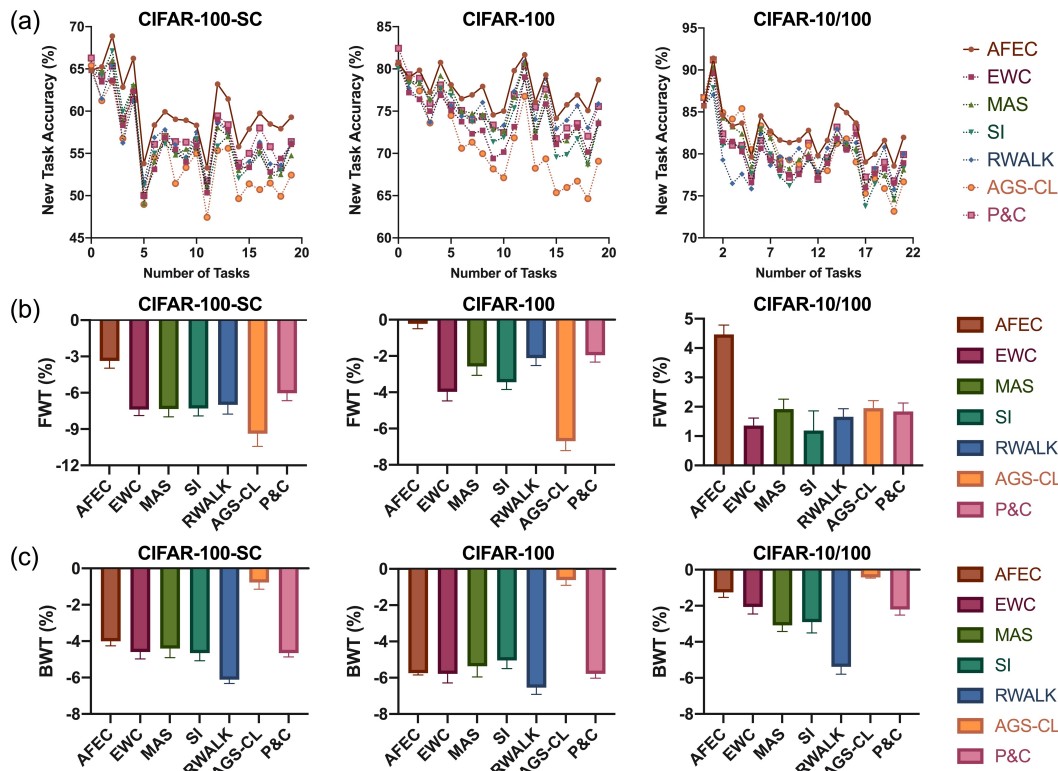

Figure 5: Knowledge transfer in continual learning. (a) The accuracy of learning each new task in continual learning. (b) Forward Transfer (FWT), which is from more negative to more positive on CIFAR-100-SC, CIFAR-100 and CIFAR-10/100. (c) Backward Transfer (BWT).

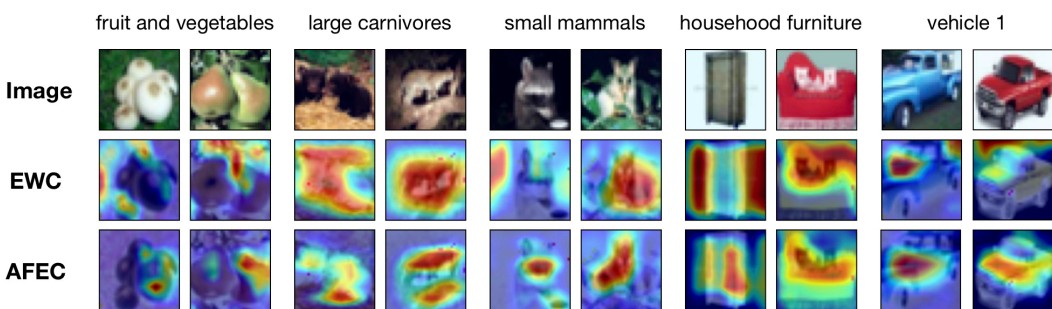

Figure 6: Visualization of predictions of the latest task after continual learning on CIFAR-100-SC. We present the results on five different random seeds, which determine five different superclasses.

in continual learning, where AFEC learns each new task much better than other baselines. Since continual learning of more tasks leads to less network resources for a new task, the overall trend of all the baselines is *declining*, indicating the necessity to improve forward knowledge transfer on the basis of overcoming catastrophic forgetting. Then we calculate forward transfer (FWT) [17], i.e., the averaged influence that learning the previous tasks has on a future task, and backward transfer (BWT) [17], i.e., the averaged influence that learning a new task has on the previous tasks (detailed in Appendix D). FWT is from more negative to more positive in CIFAR-100-SC, CIFAR-100 and CIFAR-10/100, while AFEC achieves the highest FWT among all the baselines. The BWT of AFEC is comparable as EWC, indicating that the proposed active forgetting does not cause additional catastrophic forgetting. Therefore, the performance improvement of AFEC in Table 2 is achieved by effectively improving the learning of new tasks in continual learning. In particular, AFEC achieves a much larger improvement on the learning of new tasks than P&C, which attempted to improve forward transfer of EWC through learning an additional active column. Due to the progressive

parameter isolation, although AGS-CL achieves the best BWT, its ability of learning each new task drops more rapidly than other baselines. Thus, it underperforms AFEC in Table 2.

**Visual Explanation:** To explicitly show how AFEC improves continual learning, in Fig. 6 we use Grad-CAM [27] to visualize predictions of the latest task after continual learning on CIFAR-100-SC, where FWT is more negative as discussed above. The predictions of EWC overfit the background information since it attempts to best remember the old tasks with severe negative transfer, which limits the learning of new tasks. In contrast, the visual explanation of AFEC is much more reasonable than EWC, indicating the efficacy of active forgetting to address potential negative transfer and benefit the learning of new tasks.

**Plugging-in Memory Replay:** We further implement AFEC in representative memory replay approaches in Appendix F, where we perform single-head evaluation [2]. On CIFAR-100 and ImageNet-100 datasets, we follow [9, 6] that first learn 50 classes and then continually learn the other 50 classes by 5 phases (10 classes per phase) or 10 phases (5 classes per phase), using a small memory buffer of 20 images per class. AFEC substantially boosts the performance of representative memory replay approaches such as iCaRL [21], LUCIR [9] and PODNet [6].

### 4.3 Atari Reinforcement Tasks

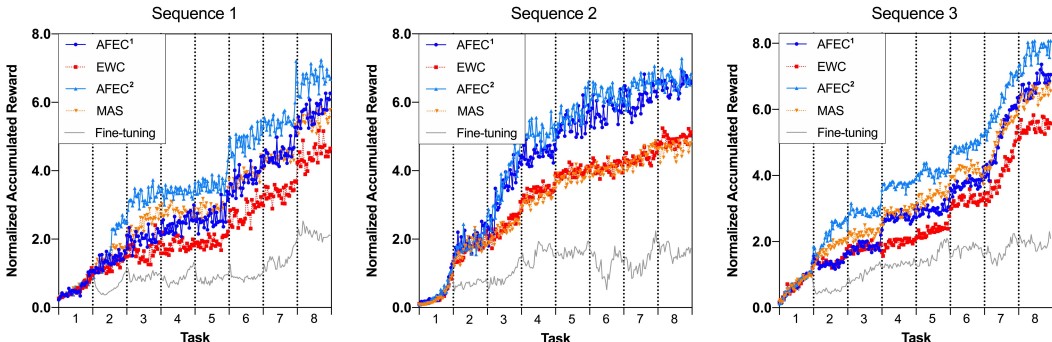

Figure 7: Continual learning of Atari reinforcement tasks. AFEC$^1$ is our method described in Sec. 3.2, while AFEC$^2$ is the adaptation of our method to MAS.

Next, we evaluate AFEC in continual learning of Atari reinforcement tasks (Atari games). We follow the implementation of [10] to sequentially learn eight randomly selected Atari games. Specifically, we applies a CNN architecture consisting of 3 convolution layers with 2 fully connected layers and identical PPO [25] for all the methods (detailed in Appendix G). The evaluation metric is the normalized accumulated reward: the evaluated rewards are normalized with the maximum reward of fine-tuning on each task, and accumulated. We present the results of three different orders of task sequence, averaged by five runs with different random initialization.

For continual learning of Atari reinforcement tasks, forward knowledge transfer is severely negative, possibly because the optimal policies of each Atari games are highly different. We first measure the normalized rewards of learning each task with a randomly initialized network, which are 2.16, 1.44 and 1.68 on the three task sequences,

Table 3: Averaged performance increase of learning each new task on Atari reinforcement tasks.

|  | Sequence 1 | Sequence 2 | Sequence 3 |
|---|---|---|---|
| AFEC$^1$ on EWC | +35.28% | +50.55% | +28.00% |
| AFEC$^2$ on MAS | +30.09% | +61.12% | +26.63% |

respectively. That is to say, the initialization learned from the old tasks results in an averaged performance decline by 53.67%, 30.66% and 40.56%, compared with random initialization. Then, we evaluate the maximum reward of learning each new task in Table 3, and the normalized accumulated reward of continual learning in Fig. 7. AFEC effectively improves the learning of new tasks and thus boosts the performance of EWC and MAS, particularly when learning more incremental tasks. AFEC also achieves a much better performance than the reproduced results of AGS-CL on its officially released code [10] (see Appendix G for an extensive analysis).

# 5 Conclusion

In this work, we draw inspirations from the biological active forgetting and propose a novel approach to mitigate potential negative transfer in continual learning. Our method achieves the SOTA performance on a variety of continual learning benchmarks through effectively improving the learning of new tasks, and boosts representative continual learning strategies in a plug-and-play way. Intriguingly, derived from active forgetting of the past with Bayesian continual learning, we obtain the algorithm that is formally consistent with the synaptic expansion and synaptic convergence (detailed Appendix A), and is functionally consistent with the advantage of biological active forgetting in memory flexibility [5]. This connection provides a potential theoretical explanation of how the underlying mechanism of biological active forgetting achieves its function of forgetting the past and continually learning conflicting experiences. We will further explore it with artificial neural networks and biological neural networks in the future.

## Limitation and Social Impact

The potential limitations of our work include three aspects: First, we propose a method to mitigate potential negative transfer in continual learning, so the efficacy of our method might be influenced by the level of negative transfer in a task sequence. Second, following [12], we assume that all the incremental tasks are independent, which might limit the application of our method to other scenarios such as continual learning of smoothly-changed data distributions. Third, our method needs to learn an additional set of parameters, resulting in more computational cost. Since our work is a fundamental research in machine learning, the negative social impacts are not obvious at this stage.

## Acknowledgements

This work was supported by NSF of China Projects (Nos. 62061136001, 61620106010, U19B2034, U181146, 62076145), Beijing NSF Project (No. JQ19016), Tsinghua-Peking Center for Life Sciences, Beijing Academy of Artificial Intelligence (BAAI), Tsinghua-Huawei Joint Research Program, a grant from Tsinghua Institute for Guo Qiang, and the NVIDIA NVAIL Program with GPU/DGX Acceleration.

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
