## A    Neural Mechanism of Biological Active Forgetting

Forgetting is an important mechanism in biological learning and memory. The biological forgetting is not simply passive, but can be actively regulated by specialized signaling pathways. An identified pathway is called Rac1 signaling pathway, where the active forgetting regulated by Rac1 signaling pathway is called Rac1-dependent active forgetting [29]. So, why do organisms evolve such a mechanism to actively forget the learned information? A study discovered that the abnormality of Rac1-dependent active forgetting results in severe defects of *memory flexibility*, where the organisms cannot effectively learn a new experience that conflicts with the old memory [5].

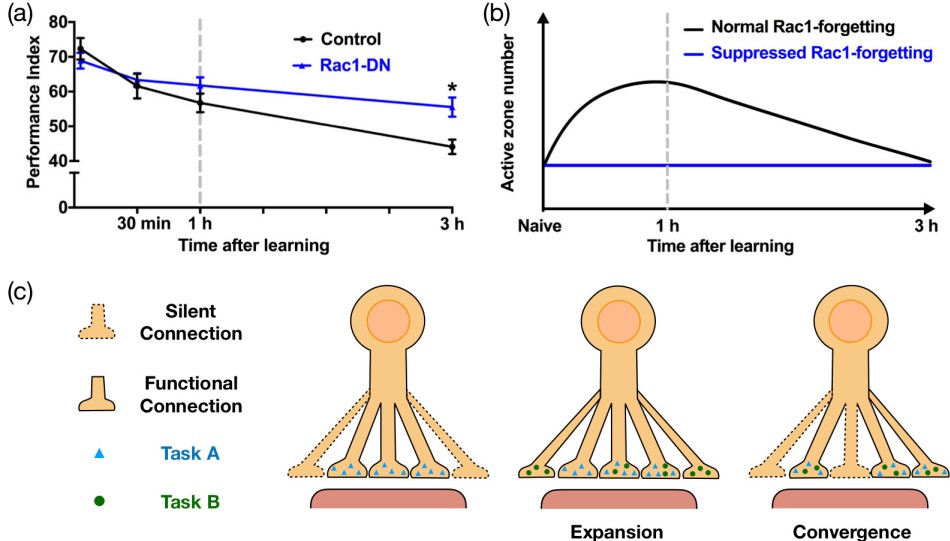

Figure 8: Summary of the mechanism underlying Rac1-dependent active forgetting at the level of synaptic structures. (a) Down-regulation of Rac1 through dominant negative overexpression (Rac1-DN) substantially slows down forgetting of the learned experience. (b) Rac1-dependent active forgetting is achieved by regulating the learning-triggered synaptic expansion and synaptic convergence. (c) A conceptual model of the learning-triggered synaptic expansion-convergence in continual learning.

However, the understanding to the neural mechanism of active forgetting is still limited. Our latest biological data in drosophila indicated that Rac1-dependent active forgetting is achieved by regulating a synaptic expansion-convergence process. Specifically, learning of a new experience triggers the increase and subsequent elimination in the number of presynaptic active zones (AZs, i.e., the site of neurotransmitter release), which is regulated by Rac1 signaling pathway (Fig. 8). After learning an aversive olfactory conditioning task, the number of AZs is significantly increased followed by elimination within the mushroom body $\gamma$ lobe where a new memory is formed (Fig. 8, a, b). The time course of AZ addition-induced elimination is in parallel with Rac1-dependent active forgetting that lasts for only hours (Fig. 8, a, b). In particular, inhibition of Rac1 and its downstream Dia specifically blocks the increase of the number rather than the size of AZs. Suppressing activity of either Rac1 or its downstream signaling pathway blocks AZ addition. Such manipulations that block AZ addition-induced elimination all prevent forgetting.[2]

The evidences above suggested that Rac1-dependent active forgetting is achieved by regulating the learning-triggered synaptic expansion-convergence, both sufficiently and necessarily. Since Rac1-dependent active forgetting is critical for organisms to continually learn a new task that conflicts with the old knowledge [5], we adapt the synaptic expansion-convergence to the scenario of continual learning (see Fig. 8, c). After learning the historical experiences (task $A$), the neural network continually learns a new experience (task $B$). The learning of task $B$ triggers the synaptic expansion, where both the expanded and the old AZs can learn the new experience. While, the subsequent synaptic convergence eliminates the AZs to the amount before learning.

---

[2]The detailed biological evidence will be published elsewhere, so we do not include them in the published version of this paper.

## B Computation Details

### B.1 Laplace Approximation

The objective of continual learning is to estimate $\theta^* = \arg\max_\theta \log p(\theta|D_A^{train}, D_B^{train})$, which can be written as:

$$\log p(\theta|D_A^{train}, D_B^{train}) = \log p(D_B^{train}|\theta) + \log p(\theta|D_A^{train}) - \log p(D_B^{train}). \qquad (11)$$

As the posterior $p(\theta|D_A^{train})$ is generally intractable (except very special cases), we must resort to approximation methods, such as the Laplace approximation [12]. If $p(\theta|D_A^{train})$ is smooth and majorly peaked around its point of maxima (i.e., $\theta_A^*$), we can approximate it with a Gaussian distribution with mean $\theta_A^*$ and variance $\sigma_A^2$. To determine $\theta_A^*$ and $\sigma_A^2$ of the Gaussian distribution, we begin with computing the second order Taylor expansion of a function $l(\theta)$ around $\theta_A^*$ as follows:

$$l(\theta) = l(\theta_A^*) + (\frac{\partial l(\theta)}{\partial\theta}|_{\theta_A^*})(\theta - \theta_A^*) + \frac{1}{2}(\theta - \theta_A^*)^{\mathrm{T}}(\frac{\partial^2 l(\theta)}{\partial\theta^2}|_{\theta_A^*})(\theta - \theta_A^*) + R_2(x), \qquad (12)$$

where $R_2(x)$ is the higher order term. Neglecting the higher order term, we have:

$$l(\theta) \approx l(\theta_A^*) + (\frac{\partial l(\theta)}{\partial\theta}|_{\theta_A^*})(\theta - \theta_A^*) + \frac{1}{2}(\theta - \theta_A^*)^{\mathrm{T}}(\frac{\partial^2 l(\theta)}{\partial\theta^2}|_{\theta_A^*})(\theta - \theta_A^*). \qquad (13)$$

Now we approximate $p(\theta|D_A^{train})$ with Eqn. (13). Noting that $\frac{\partial \log p(\theta|D_A^{train})}{\partial\theta}|_{\theta_A^*} = 0$, we have:

$$\begin{aligned}
\log p(\theta|D_A^{train}) &\approx \log p(\theta_A^*|D_A^{train}) + \frac{1}{2}(\theta - \theta_A^*)^{\mathrm{T}}(\frac{\partial^2 \log p(\theta|D_A^{train})}{\partial\theta^2}|_{\theta_A^*})(\theta - \theta_A^*) \\
&= \delta + \frac{1}{2}(\theta - \theta_A^*)^{\mathrm{T}}(\frac{\partial^2 \log p(\theta|D_A^{train})}{\partial\theta^2}|_{\theta_A^*})(\theta - \theta_A^*).
\end{aligned} \qquad (14)$$

Then, we can rewrite Eqn. (14) to obtain the Laplace approximation of $p(\theta|D_A^{train})$ as:

$$p(\theta|D_A^{train}) = \exp(\delta)\exp(-\frac{1}{2}(\theta - \theta_A^*)^{\mathrm{T}}((-\frac{\partial^2 \log p(\theta|D_A^{train})}{\partial\theta^2}|_{\theta_A^*})^{-1})^{-1}(\theta - \theta_A^*)), \qquad (15)$$

$$p(\theta|D_A^{train}) \sim N(\theta_A^*, (-\frac{\partial^2 \log p(\theta|D_A^{train})}{\partial\theta^2}|_{\theta_A^*})^{-1}). \qquad (16)$$

The variance represents the inverse of Fisher Information matrix (FIM) $F_A$, which can be approximated from the first order derivatives to avoid computing the Hessian matrix [11]:

$$\begin{aligned}
F_A &= \mathbb{E}[-\frac{\partial^2 \log p(\theta|D_A^{train})}{\partial\theta^2}|_{\theta_A^*}] \\
&\approx \mathbb{E}[(\frac{\partial \log p(\theta|D_A^{train})}{\partial\theta})(\frac{\partial \log p(\theta|D_A^{train})}{\partial\theta})|_{\theta_A^*}].
\end{aligned} \qquad (17)$$

Taking Eqn. (14) and Eqn. (11) together, we obtain the objective of EWC [12] in Eqn. (5). To address continual learning of more than two tasks, we follow [12] that averages the FIM among all the tasks ever seen for EWC and AFEC. If the network continually learns $t$ tasks, we compute the FIM of the current task as $F_t$ and update the FIM of all the old tasks as

$$F_{1:t} = ((t-1)\,F_{1:t-1} + F_t)/t. \qquad (18)$$

In Eqn. (9), the posterior $p(\theta|D_B^{train})$ can be similarly approximated as a Gaussian distribution with mean $\theta_e^*$ and variance $\sigma_e^2$. In particular, the inverse of $\sigma_e^2$ can be computed as:

$$F_e \approx \mathbb{E}[(\frac{\partial \log p(\theta|D_B^{train})}{\partial\theta})(\frac{\partial \log p(\theta|D_B^{train})}{\partial\theta})|_{\theta_e^*}]. \qquad (19)$$

## B.2 Weighted Product Distribution with Forgetting Factor

Here we prove that if two distribution $p_1(x) \sim N(\mu_1, \sigma_1^2)$, $p_2(x) \sim N(\mu_2, \sigma_2^2)$, then we can find a normalizer $Z$ that depends on $\beta$ to keep $p_m(x) = \frac{p_1(x)^{1-\beta}p_2(x)^\beta}{Z}$ following a Gaussian distribution. The probability density functions of $p_1(x)$ and $p_2(x)$ are

$$p_1(x) = \frac{1}{\sqrt{2\pi\sigma_1^2}}e^{-\frac{(x-\mu_1)^2}{2\sigma_1^2}}, \tag{20}$$

$$p_2(x) = \frac{1}{\sqrt{2\pi\sigma_2^2}}e^{-\frac{(x-\mu_2)^2}{2\sigma_2^2}}, \tag{21}$$

So we get

$$p_1(x)^{1-\beta}p_2(x)^\beta = \frac{1}{\sqrt{2\pi\sigma_1^{2(1-\beta)}\sigma_2^{2\beta}}}e^{-[\frac{(1-\beta)(x-\mu_1)^2}{2\sigma_1^2}+\frac{\beta(x-\mu_2)^2}{2\sigma_2^2}]}$$

$$= \frac{1}{\sqrt{2\pi\sigma_1^{2(1-\beta)}\sigma_2^{2\beta}}}e^{-[\frac{(x-m)^2}{2v^2}+\frac{k-m^2}{2v^2}]}. \tag{22}$$

where

$$v^2 = \frac{\sigma_1^2\sigma_2^2}{\beta\sigma_1^2 + (1-\beta)\sigma_2^2}, \tag{23}$$

$$m = \frac{(1-\beta)\sigma_2^2\mu_1 + \beta\sigma_1^2\mu_2}{\beta\sigma_1^2 + (1-\beta)\sigma_2^2}, \tag{24}$$

$$k = \frac{(1-\beta)\sigma_2^2\mu_1^2 + \beta\sigma_1^2\mu_2^2}{\beta\sigma_1^2 + (1-\beta)\sigma_2^2}. \tag{25}$$

Then we get

$$p_m(x) = \frac{p_1(x)^{1-\beta}p_2(x)^\beta}{Z} \sim N(m, v^2), \tag{26}$$

$$Z = \sqrt{\frac{v^2}{\sigma_1^{2(1-\beta)}\sigma_2^{2\beta}}}e^{-\frac{k-m^2}{2v^2}}. \tag{27}$$

## B.3 New Log Posterior of AFEC

In AFEC, the original posterior $p(\theta|D_A^{train})$ is replaced by $p_m(\theta|D_A^{train}, \beta)$ defined in Eqn. (6). Then the new log posterior $\log p(\theta|D_A^{train}, D_B^{train}, \beta)$ becomes:

$$\begin{aligned}\log p(\theta|D_A^{train}, D_B^{train}, \beta) &= \log p(D_B^{train}|\theta) + \log p_m(\theta|D_A^{train}, \beta) - \log p(D_B^{train}) \\ &= (1-\beta)[\log p(D_B^{train}|\theta) + \log p(\theta|D_A^{train}) - \log p(D_B^{train})] \\ &\quad + \beta[\log p(D_B^{train}|\theta) + \log p(\theta) - \log p(D_B^{train})] - \log Z \\ &= (1-\beta)[\log p(D_B^{train}|\theta) + \log p(\theta|D_A^{train})] + \beta\log p(\theta|D_B^{train}) \\ &\quad - \underbrace{[(1-\beta)\log p(D_B^{train}) + \log Z]}_{const.}, \end{aligned} \tag{28}$$

where $(1-\beta)\log p(D_B^{train}) + \log Z$ only depends on $\beta$ and is constant to $\theta$. Note that Eqn. (28) can be further derived as

$$\log p(\theta|D_A^{train}, D_B^{train}, \beta) = (1-\beta)\log p(\theta|D_A^{train}, D_B^{train}) + \beta\log p(\theta|D_B^{train}) + const., \tag{29}$$

$$p(\theta|D_A^{train}, D_B^{train}, \beta) = \frac{p(\theta|D_A^{train}, D_B^{train})^{(1-\beta)}p(\theta|D_B^{train})^\beta}{Z}. \tag{30}$$

As proved in Appendix B.2, the new posterior $p(\theta|D_A^{train}, D_B^{train}, \beta)$ follows a Gaussian distribution if $p(\theta|D_A^{train}, D_B^{train})$ and $p(\theta|D_B^{train})$ are both Gaussian. Therefore, we can use a Laplace approximation of it, as discussed in the main text.

## B.4 Continual Learning of More than Two Tasks

Here we discuss the scenario of continual learning of more than two tasks. First, let's consider the case of three tasks, where the neural network continually learns task $C$ from its training dataset $D_C^{train}$ after learning task $A$ and task $B$ with active forgetting. Now we use the old posterior $p(\theta|D_A^{train}, D_B^{train}, \beta_B)$ as the prior to incorporate the new task, where $\beta_B$ is the forgetting factor used to learn task $B$:

$$p(\theta|D_A^{train}, D_B^{train}, D_C^{train}) = \frac{p(D_C^{train}|\theta)\, p(\theta|D_A^{train}, D_B^{train}, \beta_B)}{p(D_C^{train})}. \tag{31}$$

To mitigate potential negative transfer to task $C$, we replace $p(\theta|D_A^{train}, D_B^{train}, \beta_B)$ that absorbs all the information of $D_A^{train}$ and $D_B^{train}$ with

$$p_m(\theta|D_A^{train}, D_B^{train}, \beta_B, \beta) = \frac{p(\theta|D_A^{train}, D_B^{train}, \beta_B)^{(1-\beta)}p(\theta)^{\beta}}{Z}. \tag{32}$$

$\beta$ is the forgetting factor to learn task $C$. $Z$ is the normalizer that depends on $\beta$, which keeps $p_m(\theta|D_A^{train}, D_B^{train}, \beta_B, \beta)$ following a Gaussian distribution if $p(\theta|D_A^{train}, D_B^{train}, \beta_B)$ and $p(\theta)$ are both Gaussian (proved in Appendix B.2). Next, we use a Laplace approximation of $p(\theta|D_A^{train}, D_B^{train}, D_C^{train}, \beta)$, and the MAP estimation is

$$\theta_{A,B,C}^* = \arg\max_{\theta} \log p(\theta|D_A^{train}, D_B^{train}, D_C^{train}, \beta)$$

$$= \arg\max_{\theta} (1-\beta)\left(\log p(D_C^{train}|\theta) + \log p(\theta|D_A^{train}, D_B^{train}, \beta_B)\right) + \beta \log p(\theta|D_C^{train}) + const.. \tag{33}$$

Then we obtain the loss function to learn the third task:

$$L_{\text{AFEC}}(\theta) = L_C(\theta) + \frac{\lambda}{2}\sum_i F_{A,B,i}(\theta_i - \theta_{A,B,i}^*)^2 + \frac{\lambda_e}{2}\sum_i F_{e,i}(\theta_i - \theta_{e,i}^*)^2, \tag{34}$$

where $L_C(\theta)$ is the loss for task $C$, $\theta_{A,B}^*$ has been obtained after learning task $A$ and task $B$, and $F_{A,B}$ is the FIM updated with Eqn. (18). $\theta_e^*$ is obtained by optimizing the expanded parameter with $L_C(\theta_e)$ and its FIM $F_e$ is calculated similarly as Eqn. (19).

Similarly, for continual learning of more tasks, where a neural network with parameter $\theta$ continually learns $T$ tasks from their task specific training datasets $D_t^{train}$ ($t = 1, 2, ..., T$), the loss function is

$$L_{\text{AFEC}}(\theta) = L_T(\theta) + \frac{\lambda}{2}\sum_i F_{1:T-1,i}(\theta_i - \theta_{1:T-1,i}^*)^2 + \frac{\lambda_e}{2}\sum_i F_{e,i}(\theta_i - \theta_{e,i}^*)^2. \tag{35}$$

To demonstrate our method more clearly, we provide a pseudocode as below:

---
**Algorithm 1** AFEC Algorithm
---
1: **Require:** $\theta$: the main network parameters; $\theta_e$: the expanded parameters; $\lambda$, $\lambda_e$: hyperparameters; $D_t^{train}$: training dataset of task $t$, $t = 1, 2, ..., T$.
2: **for** task $t = 1, 2, ..., T$ **do**
3:     // *Synaptic Expansion*
4:     Learn $\theta_e$ with $L_T(\theta_e)$ to obtain $\theta_e^*$;
5:     Calculate $F_e$ with Eqn. (19);
6:     // *Synaptic Convergence*
7:     Learn $\theta$ with Eqn. (35);
8:     Calculate $F_T$ with Eqn. (17);
9:     Update $F_{1:T}$ with Eqn. (18);
10: **end for**
---

## C  Details of Visual Classification Tasks

### C.1  Implementation

We follow the implementation of [10] for visual classification tasks with small-scale and large-scale images. For CIFAR-100-SC, CIFAR-100 and CIFAR-10/100, we use Adam optimizer with initial learning rate 0.001 to train all methods with mini-batch size of 256 for 100 epochs. For CUB-200 w/ PT, CUB-200 w/o PT and ImageNet-100, we use SGD with momentum 0.9 and initial learning rate 0.005 to train all methods with mini-batch size of 64 for 40 epochs. We make an extensive hyperparameter search of all methods and report the best performance for fair comparison. The range of hyperparameter search and the selected hyperparameter are summarized in Table 4. Due to the space limit, we include error bar (standard deviation) of the classification results in Table. 5.

### C.2  Longer Task Sequence

We further evaluate AFEC on 50-split Omniglot [10]. The averaged accuracy of the first 25 tasks is 66.45% for EWC and 84.08% for AFEC, while the averaged accuracy of all the 50 tasks is 76.53% for EWC and 83.00% for AFEC, respectively. Therefore, AFEC can still effectively improve continual learning for a much larger number of tasks.

## D  ACC, FWT and BWT

We evaluate continual learning of visual classification tasks by three metrics: averaged accuracy (AAC), forward transfer (FWT) and backward transfer (BWT) [17].

$$\text{AAC} = \frac{1}{\text{T}} \sum_{i=1}^{\text{T}} \text{A}_{\text{T},i}, \tag{36}$$

$$\text{BWT} = \frac{1}{\text{T} - 1} \sum_{i=1}^{\text{T}-1} \text{A}_{\text{T},i} - \text{A}_{i,i}, \tag{37}$$

$$\text{FWT} = \frac{1}{\text{T} - 1} \sum_{i=2}^{\text{T}} \text{A}_{i-1,i} - \bar{\text{A}}_i, \tag{38}$$

where $\text{A}_{j,k}$ is the test accuracy task $k$ after continual learning of task $j$, and $\bar{\text{A}}_k$ is the test accuracy of each task $k$ at random initialization. *Averaged accuracy* (ACC) is the averaged performance on all the tasks ever seen to evaluate the performance of both the old tasks and the new tasks.

*Forward transfer* (FWT) indicates the averaged influence that learning a task has on a future task, which can be either positive or negative. If a new task conflicts with the old tasks, negative transfer will substantially decrease the performance on the task sequence, which is a common issue for existing continual learning strategies. Since AFEC aims to improve the learning of new tasks in continual learning, FWT should reflect this advantage.

*Backward transfer* (BWT) indicates the averaged influence that learning a new task has on the old tasks. Positive BWT exists when learning of a new task increases the performance on the old tasks. On the other hand, negative BWT exists when learning of a task decreases the performance on the old tasks, which is also known as catastrophic forgetting.

## E  Adapt AFEC to Representative Weight Regularization Approaches

Regular weight regularization approaches, such as EWC [12], MAS [1], SI [35] and RWALK [2], generally add a regularization (Reg) term to penalize changes of the important parameters for the old tasks. The loss function of such methods can be written as:

$$L_{\text{Reg}}(\theta) = L_{\text{B}}(\theta) + \frac{\lambda}{2} \sum_i \xi_{A,i} (\theta_i - \theta_{A,i}^*)^2, \tag{39}$$

Table 4: Hyperparameter search for continual learning of visual classification tasks. We present the range of hyperparameter search and bold the selected hyperparameter. [1]We follow the implementation of [10] for CUB-200 w/ PT. [2]For AGS-CL [10], we make an extensive grid search of $\lambda$ and $\mu$. We follow [10] to choose $\rho$ for the variants of CIFAR-100 and CUB-200 w/ PT, and make a grid search on CUB-200 w/o PT and ImageNet-100. [3]For P&C [26], we make a grid search of the hyperparameter that controls the EWC penalty. [4]Since $\beta$ is integrated into $\lambda_e$, AFEC only needs to make a grid search of $\lambda_e$ while keeping $\lambda$ the same as the corresponding weight regularization approaches.

| Methods | Hyperparameter | CIFAR-100-SC | CIFAR-100 | CIFAR-10/100 | [1]CUB-200 w/ PT | CUB-200 w/o PT | ImageNet-100 |
|---|---|---|---|---|---|---|---|
| [2]AGS-CL [10] | $\lambda$ | 400, 800, 1600, **3200**, 6400 | 400, 800, **1600**, 3200 | 1000, 4000, **7000**, 10000 | 1.5 | 0.0001, **0.001**, 0.01, 0.1, 1, 1.5 | 0.0001, **0.001**, 0.01, 0.1, 1, 1.5, 3 |
| | $\mu$ | 5, **10**, 20, 40 | 5, **10**, 20 | 10, **20**, 40 | 0.5 | 0.001, **0.01**, 0.1, 0.5 | 0.0001, **0.001**, 0.01, 0.1, 0.5 |
| | $\rho$ | **0.3** | **0.3** | **0.2** | **0.1** | 0.05, **0.1**, 0.2 | 0.05, **0.1**, 0.2 |
| EWC [12] | $\lambda$ | 10000, 20000, **40000**, 80000 | **10000**, 20000, 40000 | 10000, **25000**, 50000, 100000 | 40 | 0.1, **1**, 5, 10, 20, 40, 80 | 40, **80**, 160, 320 |
| [3]P&C [26] | $\lambda$ | 10000, 20000, **40000**, 80000 | 10000, **20000**, 40000 | 10000, **25000**, 50000, 100000 | 40 | 0.1, **1**, 10 | 40, **80**, 160 |
| MAS [1] | $\lambda$ | 4, 8, **16**, 32 | 2, **4**, 8 | 1, 2, **5**, 10 | 0.6 | 0.001, **0.01**, 0.05, 0.5, 1.2 | 0.01, 0.03, **0.1**, 0.3, 1 |
| SI [35] | $\lambda$ | 4, **8**, 16, 32 | 2, 4, 8, **10**, 20 | 0.7, 3, **6**, 12 | 0.75 | 0.1, 0.2, **0.4**, 0.75, 1.5 | 0.3, **1**, 3, 10 |
| RWALK [2] | $\lambda$ | 64, **128**, 256 | 1, 3, **6**, 8 | 6, 12, 24, **48**, 96 | 50 | 10, **25**, 50, 100 | 0.3, 1, **3**, 10 |
| [4] w/ AFEC (ours) | $\lambda_e$ | 0.1, **1**, 10 | 0.1, **1**, 10 | 0.1, **1**, 10 | 0.1, 0.01, **0.001**, 0.0001 | 1, 0.1, **0.01**, 0.001 | 0.01, **0.001**, 0.0001 |

Table 5: Averaged accuracy (%) of all the tasks learned so far in continual learning of visual classification tasks, averaged by 5 different random seeds with error bar ($\pm$ standard deviation).

| | CIFAR-100-SC | | CIFAR-100 | | CIFAR-10/100 | | CUB-200 w/ PT | | CUB-200 w/o PT | | ImageNet-100 | |
|---|---|---|---|---|---|---|---|---|---|---|---|---|
| Methods | $A_{10}$ | $A_{20}$ | $A_{10}$ | $A_{20}$ | $A_2$ | $A_{2+20}$ | $A_5$ | $A_{10}$ | $A_5$ | $A_{10}$ | $A_5$ | $A_{10}$ |
| Fine-tuning | 32.58 $\pm 1.41$ | 28.40 $\pm 0.78$ | 40.92 $\pm 4.33$ | 33.53 $\pm 3.25$ | 78.96 $\pm 1.74$ | 37.81 $\pm 0.99$ | 78.75 $\pm 0.99$ | 78.13 $\pm 0.61$ | 31.91 $\pm 2.38$ | 39.82 $\pm 2.26$ | 50.56 $\pm 1.97$ | 44.80 $\pm 2.65$ |
| P&C [26] | 53.48 $\pm 2.79$ | 52.88 $\pm 1.68$ | 70.10 $\pm 1.22$ | 70.21 $\pm 1.22$ | 86.72 $\pm 1.33$ | 78.29 $\pm 0.74$ | 81.42 $\pm 1.72$ | 81.74 $\pm 1.10$ | 33.88 $\pm 4.48$ | 42.79 $\pm 3.29$ | 76.44 $\pm 1.65$ | 74.38 $\pm 1.61$ |
| AGS-CL [10] | 55.19 $\pm 2.09$ | 53.19 $\pm 1.04$ | 71.24 $\pm 0.77$ | 69.99 $\pm 1.06$ | 86.27 $\pm 0.79$ | 80.42 $\pm 0.69$ | 82.30 $\pm 0.38$ | 81.84 $\pm 0.51$ | 32.69 $\pm 3.45$ | 40.73 $\pm 2.82$ | 51.48 $\pm 1.74$ | 47.20 $\pm 2.05$ |
| EWC [12] | 52.25 $\pm 2.99$ | 51.74 $\pm 1.74$ | 68.72 $\pm 0.24$ | 69.18 $\pm 0.69$ | 85.07 $\pm 0.84$ | 77.75 $\pm 0.57$ | 81.37 $\pm 0.28$ | 80.92 $\pm 1.04$ | 32.90 $\pm 2.98$ | 42.29 $\pm 2.34$ | 76.12 $\pm 2.08$ | 73.82 $\pm 1.46$ |
| AFEC (ours) | 56.28 $\pm 3.27$ | 55.24 $\pm 1.61$ | 72.36 $\pm 1.23$ | 72.29 $\pm 1.07$ | 86.87 $\pm 0.78$ | 81.25 $\pm 0.23$ | 83.65 $\pm 0.55$ | 82.04 $\pm 0.77$ | 34.36 $\pm 4.39$ | 43.05 $\pm 3.00$ | 77.64 $\pm 1.80$ | 75.46 $\pm 1.30$ |
| MAS [1] | 52.76 $\pm 2.85$ | 52.18 $\pm 2.22$ | 67.60 $\pm 1.85$ | 69.41 $\pm 1.27$ | 84.97 $\pm 0.51$ | 77.39 $\pm 1.03$ | 79.98 $\pm 1.41$ | 79.67 $\pm 0.77$ | 31.68 $\pm 2.37$ | 42.56 $\pm 1.84$ | 75.48 $\pm 0.66$ | 74.72 $\pm 0.79$ |
| *w/ AFEC (ours)* | 55.26 $\pm 4.14$ | 54.89 $\pm 2.23$ | 69.57 $\pm 1.73$ | 71.20 $\pm 0.70$ | 86.21 $\pm 1.24$ | 80.01 $\pm 0.51$ | 82.77 $\pm 0.32$ | 81.31 $\pm 0.25$ | 34.08 $\pm 3.80$ | 42.93 $\pm 3.51$ | 75.64 $\pm 0.94$ | 75.66 $\pm 1.33$ |
| SI [35] | 52.20 $\pm 4.37$ | 51.97 $\pm 2.07$ | 68.72 $\pm 1.11$ | 69.21 $\pm 0.77$ | 85.00 $\pm 2.52$ | 76.69 $\pm 2.11$ | 80.14 $\pm 0.88$ | 80.21 $\pm 0.89$ | 33.08 $\pm 4.05$ | 42.03 $\pm 3.06$ | 73.52 $\pm 1.35$ | 72.97 $\pm 1.85$ |
| *w/ AFEC (ours)* | 55.25 $\pm 3.43$ | 53.90 $\pm 2.31$ | 69.34 $\pm 1.87$ | 70.13 $\pm 1.36$ | 85.71 $\pm 1.08$ | 78.49 $\pm 0.89$ | 83.06 $\pm 0.82$ | 81.88 $\pm 0.73$ | 34.04 $\pm 3.40$ | 43.20 $\pm 2.50$ | 75.72 $\pm 1.06$ | 74.14 $\pm 1.70$ |
| RWALK [2] | 50.51 $\pm 4.53$ | 49.62 $\pm 3.28$ | 66.02 $\pm 1.89$ | 66.90 $\pm 0.29$ | 85.59 $\pm 1.31$ | 73.64 $\pm 1.53$ | 80.81 $\pm 0.90$ | 80.58 $\pm 0.83$ | 32.56 $\pm 3.76$ | 41.94 $\pm 2.35$ | 73.24 $\pm 1.45$ | 73.22 $\pm 1.14$ |
| *w/ AFEC (ours)* | 52.62 $\pm 2.61$ | 51.76 $\pm 1.72$ | 68.50 $\pm 1.80$ | 69.12 $\pm 0.96$ | 86.12 $\pm 0.94$ | 77.16 $\pm 0.66$ | 83.24 $\pm 0.46$ | 81.95 $\pm 0.41$ | 33.35 $\pm 2.44$ | 42.95 $\pm 1.59$ | 74.64 $\pm 1.38$ | 73.86 $\pm 1.54$ |

where $\lambda$ is the hyperparameter that explicitly controls the strength to remember task $A$, and $\xi_{A,i}$ indicates the "importance" of parameter $i$ to task $A$.

Through plugging-in the regularization term of active forgetting, AFEC can be naturally adapted to regular weight regularization approaches. Here we consider a simple adaptation, and validate its effectiveness in Table 2:

$$L_{\text{Reg w/ AFEC}}(\theta) = L_{\text{B}}(\theta) + \frac{\lambda}{2} \sum_i \xi_{A,i}(\theta_i - \theta_{A,i}^*)^2 + \frac{\lambda_e}{2} \sum_i F_{e,i}(\theta_i - \theta_{e,i}^*)^2], \qquad (40)$$

where we learn the expanded parameters $\theta_e$ with $L_{\text{B}}(\theta_e)$ to obtain $\theta_e^*$, and calculate $F_e$ with Eqn. (19).

## F  Adapt AFEC to Representative Memory Replay Approaches

Here we relax the restriction of accessing to old training data, and plug AFEC in representative memory replay approaches such as iCaRL [21], LUCIR [9] and POD-Net [6] with single-head evaluation [2]. We follow the setting widely-used in the above memory replay approaches that the neural network first learns 50 classes and then continually learns the other 50 classes by 5 phases (10 classes per phase) or 10 phases (5 classes per phase) with a small memory buffer of 20 images per class [21, 9, 6]. We implement AFEC in the officially released code of corresponding methods for fair comparison. For CIFAR-100, we use ResNet32 and train each model for 160 epochs with

Table 6: Plugging AFEC in representative memory replay approaches with their officially released codes. We present the averaged incremental accuracy (%) on CIFAR-100 and ImageNet-100.

| | CIFAR-100 | | ImageNet-100 | |
|---|---|---|---|---|
| Methods | 5-phase | 10-phase | 5-phase | 10-phase |
| iCaRL [21] | 57.12 | 52.66 | 65.44 | 59.88 |
| w/ AFEC (ours) | 62.76 $\pm 0.52$ | 59.00 $\pm 0.72$ | 70.75 $\pm 1.12$ | 65.62 $\pm 0.78$ |
| LUCIR [9] | 63.17 | 60.14 | 70.84 | 68.32 |
| w/ AFEC (ours) | 64.47 $\pm 0.46$ | 62.26 $\pm 0.29$ | 73.38 $\pm 0.78$ | 70.20 $\pm 0.84$ |
| PODNet [6] | 64.83 | 63.19 | 75.54 | 74.33 |
| w/ AFEC (ours) | 65.86 $\pm 0.75$ | 63.79 $\pm 0.86$ | 76.90 $\pm 0.82$ | 75.80 $\pm 0.90$ |

minibatch size of 128 and weight decay of $5 \times 10^{-4}$. For ImageNet-100, we use ResNet18 and train each model for 90 epochs with minibatch size of 128 and weight decay of $1 \times 10^{-4}$. For all the datasets, we use a SGD optimizer with momentum of 0.9, initial learning rate of 0.1 and cosine annealing scheduling. As shown in Table 6, AFEC substantially boosts the performance of representative memory replay approaches such as iCaRL [21], LUCIR [9] and PODNet [6].

## G  Details of Atari Reinforcement Tasks

### G.1  Implementation

The officially released code of [10] provided the implementations of EWC, AGS-CL and fine-tuning. We reproduce the above baselines, and implement MAS, AFEC[1] and AFEC[2] on it with the same hyperparameters of PPO [25] and training details. Specifically, we use Adam optimizer with the initial learning rate of 0.0003 and evaluate the normalized accumulated reward of all the tasks ever seen for 30 times during training each task. Also, we follow [10] to search the hyperparameters of AGS-CL ($\lambda = 100, 1000$; $\mu = 0.1, 0.125$), EWC ($\lambda = 10000, 25000, 100000$) and MAS ($\lambda = 1, 10$).

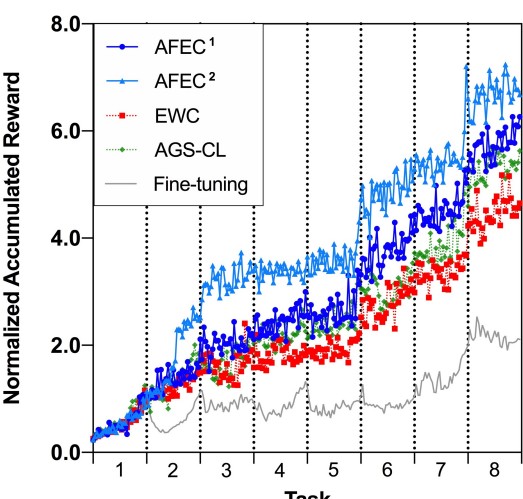

Figure 9: We use the same hyperparameters as [10] for Sequence 1: $\lambda = 100000$ for EWC, $\lambda = 100$, $\mu = 0.1$ for AGS-CL. We use $\lambda = 100000$, $\lambda_e = 100$ for AFEC[1] and $\lambda = 10$, $\lambda_e = 100$ for AFEC[2].

We observe that the normalized rewards obtained in continual learning are highly unstable in different runs and random seeds for all the baselines, possibly because the optimal policies for Atari games are highly different from each other, which results in severe negative transfer. Thus, we average the performance for five runs with different random seeds to acquire consistent results.[3] Also, we evaluate three orders of the task sequence as below:

Sequence 1 (the original task order used in [10]): StarGunner - Boxing - VideoPinball - Crazyclimber - Gopher - Robotank - DemonAttack - NameThisGame

Sequence 2: DemonAttack - Robotank - Boxing - NameThisGame - StarGunner - Gopher - VideoPinball - Crazyclimber

Sequence 3: Crazyclimber - Robotank - Gopher - NameThisGame - DemonAttack - StarGunner - Boxing - VideoPinball

### G.2  Reproduced Results of AGS-CL

However, the officially released code cannot reproduce the reported performance of AGS-CL in [10]. For the reported performance in [10], the normalized rewards of AGS-CL are significantly higher than EWC on Task 1 and Task 7, while are comparable with or slightly higher than EWC on the other six tasks. Thus, the accumulated normalized reward of AGS-CL significantly outperforms EWC in [10]. By contrast, the officially released code can only reproduce the results of AGS-CL and EWC on the other six tasks rather than on Task 1 and Task 7. In other words, the advantage of AGS-CL on Task 1 and Task 7 is far less significant than the reported performance. Therefore, the reproduced normalized accumulated reward of AGS-CL only slightly outperforms EWC, as shown in Fig. 9. Here we compare the reported and the reproduced performance on Task 1 and Task 7 in Fig. 10, and provide an extensive analysis to demonstrate why the reported performance *cannot* be reproduced.

(1) Abnormal Results of AGS-CL on Task 1 and Task 7:

---

[3][10] did not mention how many times they run and average the Atari experiments.

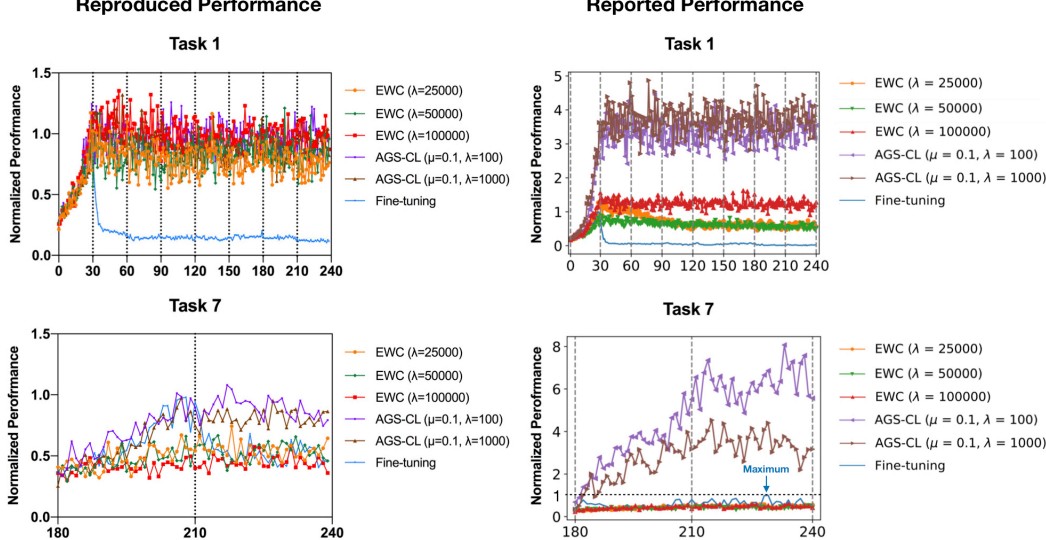

Figure 10: Comparison of the reproduced performance and the reported performance on Task 1 and Task 7. The maximum reward of fine-tuning on Task 7 is labeled in the reported performance.

As shown in Fig. 10, the reported performance of AGS-CL are $\sim 4$ and $\sim 7$ on Task 1 and Task 7, which means the obtained rewards of AGS-CL are $\sim 4$ times and $\sim 7$ times larger than the maximum rewards of fine-tuning on Task 1 and Task 7, respectively.[4] However, AGS-CL is a method that progressively isolates dedicated parameter subspace for the old tasks and randomly initializes the remaining part of the network for the new tasks [10]. The performance of AGS-CL on Task 1 should not exceed the maximum performance of fine-tuning, since they both learn the first task on the same randomly initialized network, consistent with our reproduced results. In other words, the normalized reward of AGS-CL on Task 1 should be generally less than or around 1,[5] so the reported performance ($\sim 4$) is abnormal.

On the other hand, the initialization might result in different abilities of AGS-CL and fine-tuning to learn Task 7, since fine-tuning inherits the initialization after learning the previous tasks sequentially. To evaluate the effect of initialization, we use a randomly initialized network only to learn Task 7, and normalize the obtained reward with the maximum reward of fine-tuning. We call this simple baseline as "Naive", which should be the *upperbound* performance of AGS-CL on Task 7 because the network capacity of Naive is much larger than that of AGS-CL. However, the reported performance of AGS-CL is much higher than Naive on Task 7: the maximum performance of Naive is 3.79, while AGS-CL is around 7. The analysis above is validated by the reproduced performance of AGS-CL, which is much lower than both Naive and the reported performance on Task 7.

We notice that [10] interpreted the outstanding performance of AGS-CL on Task 1 and Task 7 as the $\sim 4$ times and $\sim 7$ times higher "plasticity" than fine-tuning. However, since the "plasticity" is calculated similarly as the normalized reward, i.e., the obtained reward normalized by the performance of fine-tuning, this metric cannot provide additional information to explain the abnormal results.

(2) Abnormal Results of Fine-tuning on Task 7:

We observe that the reported performance of fine-tuning cannot effectively learn Task 7 (the window of learning Task 7 is 180-210). Specifically, the normalized reward of fine-tuning achieves a local peak at the beginning of learning Task 7 but quickly decreases to almost zero, and then becomes highly dynamic when learning Task 8. It is abnormal that the maximum reward of fine-tuning on Task 7 is achieved in the period of learning Task 8 (the window of learning Task 8 is 210-240), since learning a new task typically results in severe catastrophic forgetting of the old tasks for fine-tuning. From running the officially released code, we observe that although in very rare cases ($\sim 5\%$) all the

---

[4]The reported normalized rewards of AGS-CL on the other six tasks are generally less than or around 1, and are comparable with or slightly higher than EWC.

[5]The normalized reward might be slightly higher than 1 due to the stochasticity of training.

baselines may not be able to learn a task well in continual learning (i.e., the performance is highly dynamic around zero), all of them can effectively learn Task 7 through averaging the performance for five runs. The abnormal result of fine-tuning on Task 7 might be the cause of the abnormal result of AGS-CL due to the too small normalizer.

(3) Abnormal Results of EWC on Task 1:

Although the reported performance of EWC is broadly consistent with the reproduced one, the result on Task 1 is abnormal. As analysed in the above issue (1), EWC with different $\lambda$ should perform comparably when learning Task 1, since all of them use the same randomly initialized network *without* extra regularization on the first task while the regularizer is implemented when learning the second task. However, it can be clearly seen that the reported performance of EWC with $\lambda = 50000$ is significantly lower than the one with $\lambda = 25000$ and $\lambda = 100000$. We carefully check the code of EWC to make sure there are no extra bugs. Then we reproduce the performance of EWC and implement AFEC on it, where the reproduced results do not have the above issue.

We emailed the AGS-CL [10] authors with the reproduced results and the analysis above, but without reply yet. Therefore, we only present the results of AFEC, MAS, EWC and fine-tuning in the main text, and present the reproduced results of AGS-CL in Fig. 9, where both AFEC[1] and AFEC[2] substantially outperform AGS-CL.

## H  License of Datasets

Four public datasets are used in this paper, including CIFAR-10 [13], CIFAR-100 [13], CUB-200-2011 [31] and ImageNet [23]. We search the license of the four datasets on *paperswithcode*. The licenses of CIFAR-10, CIFAR-100 and CUB-200-2011 are "unknown".

The license of ImageNet is "Custom (research, non-commercial)": ImageNet is an image database organized according to the WordNet hierarchy (currently only the nouns), in which each node of the hierarchy is depicted by hundreds and thousands of images. The project has been instrumental in advancing computer vision and deep learning research. The data is available for free to researchers for non-commercial use.

## I  Computation Resource

Our GPU type is NVIDIA GeForce GTX 1080 Ti for all the experiments. We approximate the total amount of computation to produce the results presented in this manuscript: around 59 hours for CIFAR-10 regression tasks; around 2856 hours for visual classification tasks; around 2520 hours for Atari reinforcement tasks.