# OpenReview forum: "AFEC: Active Forgetting of Negative Transfer in Continual Learning"
_NeurIPS.cc/2021/Conference — NeurIPS 2021 Poster_

### Official Review · Reviewer_we6u · 2021-07-15

**Rating:** 6
**Confidence:** 4

**Summary:**

This paper proposes a method for continual learning that aims to mitigate the problem of catastrophic forgetting by actively forgetting knowledge from previous tasks. The authors use a Bayesian continual learning framework to design their method. AFEC relies on identifying a set of expanded network parameters which are specific to the new task. The method incorporates a term in the loss function that considers the merge between these parameters and the network parameters being updated for the current task, therefore encouraging active forgetting. Catastrophic forgetting is also controlled by including a term that merges between parameters for old tasks and network parameters for the current task (i.e. the EWC term). Experimental results are reported for datasets on regression, classification and reinforcement learning tasks. These results show some advantages of the method in particular in the forward transfer setting.

**Limitations And Societal Impact:**

I would suggest the authors to explicitly mention/point to limitations in terms of complexity of tasks (which implies a larger set of parameters) and number of tasks (which increases the challenge for balancing active forgetting and knowledge retention of old tasks).

**Main Review:**

Strengths of this paper are:
- The paper is very well-written, well-structured, and easy to follow.
- The method explores a new avenue which has been rather ignored in previous continual learning research, in terms of promoting forgetting when appropriate. I believe this is the main strength and novelty of the proposed approach.
- The contribution is technically sound, and the claims are reasonably supported by the experimental results in a variety of tasks (regression, classification, reinforcement learning)

Weaknesses of this paper are:
- In the experimental results reported in Section 4.2, it is not clear whether the 5 runs involved 5 different task orders or just 5 different random seeds. This is important as the order of tasks has a direct influence in forward and backward transfer – i.e. learning easier tasks first and then hard ones may give different results than learning hard tasks first and then easier ones.
- Although it is explicitly mentioned that the set of expanded parameters does not need to be stored for later use, the computational cost of solving Eq. 10 is not mentioned. In particular, does this increase with the size of the network, for example for a rather large number of tasks?
- Although I appreciate the analogy with biological active forgetting, a clear difference I see of the proposed approach vs. a biological system is the number of tasks. How would AFEC work well for a rather large number of tasks (e.g. hundreds, thousands)? I would believe the trade-off between actively forgetting and retaining knowledge of old tasks would be increasingly challenging with an increasing number of tasks.
- The use of some language could be improved. For example, the word “significantly” is used in several parts of the paper, although no tests for statistical significance are reported. There is also a typo at the end of the first paragraph of Section 2 (“which toke advantages… ”).


**Time Spent Reviewing:**

3

---

> ### Author Response · Authors · 2021-08-08
> **Author Response**
>
> Thank you for your valuable comments.
>
> ***Q1: The experimental results in Section 4.2:***
>
> A1: In fact, all the experimental results reported in our paper were averaged by 5 different task orders with different random seeds. We will make it clearer in the final version.
>
> ***Q2: The computational cost of solving Eq (10):***
>
> A2: Since we need to train the expanded parameters $\theta_e$ to solve Eq (10), the computational cost of AFEC is exactly twice of only training the main network parameters. We will add it in the final version.
>
> ***Q3: How would AFEC work well for a rather large number of tasks?***
>
> A3: Thanks. First, due to the limited size of representative visual classification datasets and following their commonly-used splits for continual learning, in Table 2 we reported the averaged accuracy of 10 and 20 tasks for CIFAR100-SC and CIFAR100, and 5 and 10 tasks for CUB-200 and ImageNet-100. When learning more tasks, the improvement of AFEC on average accuracy of all the tasks ever seen is roughly similar or only slightly reduced, compared with its ablation.
>
> Second, we further evaluate AFEC on 50-split Omniglot, where each alphabet with several classes is defined as a classification task. The averaged accuracy of 25 tasks is 66.45\% for EWC and 84.08\% for AFEC, while the averaged accuracy of 50 tasks is 76.53\% for EWC and 83.00\% for AFEC. Therefore, AFEC can still effectively improve continual learning for a much larger number of tasks. In particular, the stronger ability of learning new tasks enables AFEC to more quickly learn a useful model from incremental data, so it achieves a strong performance at 25 tasks although slightly drops at 50 tasks due to the difficulty of inferring more tasks. By contrast, the performance of EWC is largely limited by the small amount of training data in Omniglot, so it gradually improves with more tasks and more training data.
>
> We will clarify it in the final version, and will explore the trade-off between active forgetting and retaining old knowledge in further work.
>
> ***Q4: The use of some language could be improved:***
>
> A4: Thanks. In fact, we report all the experimental results with standard deviance and the number of runs (please see Table 1, Appendix D Table 5 and Appendix G Table 6) for significance analysis. For example, by student t-test of the accuracy, the two-tailed P-value between AFEC and EWC is both less than 0.0001 for 10-task CIFAR10 regression with VGG11BN and ResNet10, and 0.0108, 0.0006, and less than 0.0001 for 20-task CIFAR100-SC, 20-task CIFAR100 and 22-task CIFAR10/100 classification, respectively. By conventional criteria (i.e. P-value < 0.05), the difference above is considered to be statistically significant. We will improve the use of language and add a full significance analysis in the final version.

---

> > ### Comment · Reviewer_we6u · 2021-08-25
> > **Authors response**
> >
> > Thanks to the authors for their responses and additional information provided. Please make sure you incorporate any additional information to the paper/supplementary material.
> >
> > Having read all other reviews and comments, I would like to keep my rating unchanged.

---

> > > ### Author Response · Authors · 2021-08-25
> > > **Thanks for the feedback**
> > >
> > > Thank you very much for the support! We'll incorporate the additional information into the revision.

---

### Official Review · Reviewer_kQj5 · 2021-07-16

**Rating:** 6
**Confidence:** 5

**Summary:**

The paper proposed a continual learning method that attempts to promote better forward transfer among similar tasks. Towards this, the authors attempt to suppress the negative knowledge, the knowledge that impedes learning a new task, from the previous task while learning a new task. This is done by forgetting such knowledge from the previous task without increasing the forgetting catastrophically. The authors claim that such “active forgetting” is biologically inspired. Similar to the Elastic Weight Consolidation (EWC), the authors derive the learning objective, by applying the laplace approximation to the posterior of a new task. The only difference is that the posterior of the previous task is replaced by a weighted product distribution, where a weight of ‘\beta’ is introduced to capture active forgetting. This change allowed them to derive a new regularization penalty whereby a separate model is trained for a new task and then subsequently merged with that of the previous one. It is in this merging that the negative knowledge of the previous task is eliminated. Experimentally, the authors show strong performance on regression, classification and RL tasks. The authors further show that their method can be added to other regularization- and memory-based methods to improve their performance.


**Main Review:**

Positives:

- Overall the paper is well-written and intuitively easy to understand.
- Experiments are conducted thoroughly across various domains.

Negatives:

- Details of the method: It seems that some implementation details of the method are missing and, in fact, difficult to discern from the main learning objective (Eq. 10). Do the authors first train the separate network to obtain the optimal parameters for the new task $\theta_e^*$, then compute $F_e$ for this optimal model, and finally use Eq. 10 to merge the knowledge from the two tasks/ models? If not, then how is the $\theta_e^*$ and $F_e$ are obtained? If yes, then would there be an additional $L_B(\theta)$ term in objective?

- Progress and Compress (Schwarz et al.): It seems that conceptually the method is very similar to the progress and compress (P&C) where a new “active” column/ network is learned for a new task, that is later distilled to the unified model of the previous task. The authors of the present work seem to do the same thing but, of course, the mechanism of merging is different: whereas in P&C knowledge distillation is used, AFEC (present work) uses an additional EWC-type loss. I would want the authors to compare against P&C both theoretically and experimentally.

- Short term measure of negative knowledge: It seems that AFEC takes a short term view of negative knowledge (i.e.) the knowledge that is negative for only the current task. I can imagine scenarios where an orthogonal current task would force the network to remove most of the knowledge from previous task(s), because it was deemed negative for the current task, even though it would have been beneficial for the future tasks. What is the take of the authors on that?

--------------------------------------------
Post-rebuttal:

I quite likes how objectively the authors responded to the review. While the two-step training for each new task has its drawbacks, I overall quite like the paper. The authors response addressed some of my concerns. Therefore, I am increasing my score to 6.

**Time Spent Reviewing:**

4

---

> ### Author Response · Authors · 2021-08-08
> **Author Response**
>
> Thank you for your valuable comments.
>
> ***Q1: Details of the method:***
>
> A1: Thanks. Yes, we first train a separate network to obtain the optimal parameters for the new task and then compute its Fisher Information matrix, and finally use Eq (10) to merge the two models. We defined $L_B(\theta_e)$ and use it to obtain the optimal parameters (please see line 148). We will move $L_B(\theta_e)$ to our objective and add a pseudo code of AFEC to make the details clearer in the final version.
>
> ***Q2: Comparison with Progress and Compress (P\&C):***
>
> A2: Thanks for the valuable related work. Here we compare AFEC and P\&C both theoretically and empirically, and demonstrate the advantage and novelty of our paper.
>
> On the theoretical side: P\&C was motivated by the assumption that knowledge transfer from the old tasks to each new task is generally positive. To improve the positive transfer from the past, P\&C learns an active column network using layerwise adaptors, which is ''an idea borrowed from Progressive Nets''. P\&C also proposed an online EWC strategy to *mitigate catastrophic forgetting* of the unified model when distilling knowledge from the active column.
>
> By contrast, we propose that the knowledge transfer from the past to each new task might be either positive or negative (e.g., visual classification), or even severely negative in some extreme scenarios (e.g., CIFAR10 regression, Atari games), validated by extensive experiments. To mitigate potential negative transfer in continual learning, which is under addressed but highly nontrivial, we draw inspirations from biological active forgetting to actively forget the posterior distribution about the past in the framework of Bayesian continual learning, and derive the loss function in Eq (10). Therefore, the novel design in AFEC, i.e. the third term in Eq (10), is to *promote active forgetting* of the main network, which is *opposite* to P\&C due to the different motivation.
>
> In addition, the design of AFEC in Eq (10) not only provides an effective approach for continual learning, but also is to test an idea that if active forgetting of the old experience with proper theoretical basis can derive an algorithm that is formally consistent with the synaptic expansion and synaptic convergence, which is the underlying mechanism of active forgetting in biological neural networks. It provides a potential theoretical explanation for an important yet under addressed neuroscience problem: how the underlying mechanism of biological active forgetting achieves its function of forgetting the past and continually learning conflicting experiences [1-3].
>
> On the empirical side: The experimental setting of P\&C is to learn the task sequence in a *recurring* fashion, which means the past tasks can be learned again. By contrast, we follow the *pure* continual learning setting, where the past tasks cannot be learned again, but the averaged accuracy (for regression and classification) or average rewards (for reinforcement learning) are evaluated for all tasks ever seen. As analyzed by [4], the pure continual learning setting is more challenging yet realistic than the recurring setting used in P\&C.
>
> We will include the above analysis in the final version.
>
> ***Q3: Short term measure of negative knowledge:***
>
> A3: Thanks for the inspiring question.
> First, although AFEC mitigates negative transfer to each new task, we empirically validate that AFEC also brings longer-term benefits to continual learning. As shown in Table 1 for regression and Table 2 for classification, the improvement of AFEC on average accuracy of all the tasks ever seen is roughly similar or only slightly reduced when learning more tasks, compared with its ablation.
>
> Second, a similar scenario as Q3 has been evaluated in our regression experiment. As shown in Fig. 3, we can design tasks with severe negative transfers between each other (e.g., Task A and Task B) or benefit each other (e.g., Task A and ''*Transfer*'', refer to as Task A'). In Table 1, the Transfer experiment is to continually learn Task A -> Task B -> Task A', where the performance of AFEC on Task A' is substantially better than EWC, while EWC is even worse than sequential fine-tuning.
> Thus, if Task 1 severely interferes with Task 2 but benefits Task 3, to learn Task 2 well you have to largely remove the knowledge of Task 1 (e.g., for EWC you cannot use a very high penalty). Then, it's better to actively forget Task 1 to create space for Task 2 than directly learning Task 2 with ''passive'' catastrophic forgetting of Task 1, because the later strategy will more severely decline the potential positive transfer from Task 1 to Task 3.
>
> We will add this discussion in the final version.
>
>
>
>
> [1] Dong, T., He, J., et al. (2016). Inability to activate Rac1-dependent forgetting contributes to behavioral inflexibility in mutants of multiple autism-risk genes. PNAS, 113(27), 7644-7649.
>
> [2] Davis, R. L., \& Zhong, Y. (2017). The biology of forgetting—a perspective. Neuron, 95(3), 490-503.
>
> [3] Zhang, X., Li, Q., et al. (2018). Active protection: Learning-activated Raf/MAPK activity protects labile memory from Rac1-independent forgetting. Neuron, 98(1), 142-155.
>
> [4] Jung, S., Ahn, H., et al. (2020). Continual learning with node-importance based adaptive group sparse regularization. NeurIPS 2020.

---

> > ### Comment · Area_Chair_eCaL · 2021-08-24
> > **P&C and interferece**
> >
> > I think P&C does discuss the scenario of interference, and actually the method was initially proposed with forward transfer in mind and graceful forgetting. That said, I agree that the extent to which it empirically checks this things is somewhat limited. And the neuroscience argument is a good one. I do think however P&C would be a great baseline .. though realize it might be hard to do in such a short time frame.

---

> > > ### Author Response · Authors · 2021-08-24
> > > **Author Response**
> > >
> > > Thank you very much for your valuable feedback.
> > >
> > > We agree that P&C is an important related work, which assumed that knowledge transfer from the old tasks is generally *positive* but alleviating catastrophic forgetting will interfere with it. In contrast, we demonstrate that knowledge transfer from the past might be either positive or negative, while the potential *negative* transfer is the one that interferes with alleviating catastrophic forgetting. We’ll make this more explicit.
> > >
> > > As for the empirical comparison, thank you very much for the understanding on the hardness to implement P&C in a short time! Indeed, since P&C did not release their code or include a pseudo code in the original paper, it is difficult to implement and fairly evaluate P&C in such a short time frame. We will try our best to reproduce P&C, and extensively compare with it in the final version. Here, we would also like to highlight the key difference between the experimental settings: P&C *recurrently* learned the past tasks again and again, while we follow the pure continual learning setting where the past tasks cannot be learned again.

---

> > > ### Author Response · Authors · 2021-08-26
> > > **Re: P&C and interferece**
> > >
> > > Thank you very much for your valuable feedback.
> > >
> > > We have tried our best to reproduce P&C in the pure continual learning setting. Then we extensively compare P&C, EWC and AFEC (ours) in visual classification benchmarks of CIFAR100-SC, CIFAR100 and CIFAR10/100 used in our paper. Please refer to the comment entitled '**Compare with P&C in Pure Continual Learning Setting**'.

---

### Official Review · Reviewer_WUG1 · 2021-07-16

**Rating:** 6
**Confidence:** 4

**Summary:**

This paper investigates how to avoid negative transfer in Bayesian continual learning setting.
The main contribution is the proposition of an active forgetting mechanisms inspired by biological neural networks for mitigating negative transfer in CL.
The experiments on CL benchmarks have been conducted to validate the performances of the proposed method.


**Limitations And Societal Impact:**

Yes

**Main Review:**

Originality:
  This paper proposes a novel approach -- Active Forgetting with synaptic Expansion-Convergence (AFEC), for reducing the interferes of knowledge from past learning. The method can be a plug-and-play to a CL model for boosting its performance.
  Relationship to previous works has been explained and relevant literature has been cited appropriately.

Quality:
  The work is relevant to audience at Neurips and provides reasonable details and code for reproducibility.
  The authors have conducted experiments on several CL benchmarks to evaluate the effectiveness of the proposed method, and demonstrated SOTA performance.

Clarity:
  The paper is overall clearly written and the method is adequately described.
It provides reasonable details and code for reproducibility.

Significance:
  The improvement of the proposed method is significant on several CL tasks including regression, classification, and RL.

[After Rebuttal]
After reading other reviewers' comments and the authors responses, I'd like to keep my rating 6 because the authors have addressed an important question in CL and most reviewers' concerns in their responses. They also provide enough details/codes for readers to reproduce the results.

**Time Spent Reviewing:**

4 hr

---

> ### Author Response · Authors · 2021-08-08
> **Author Response**
>
> Thank you for your positive comments. We will further improve our paper in the final version.

---

> > ### Comment · Reviewer_WUG1 · 2021-08-25
> > **Re: Discussion**
> >
> > After reading other reviewers' comments and the authors responses, I'd like to keep my rating 6 because the authors have addressed an important question in CL and most reviewers' concerns in their responses.

---

> > > ### Author Response · Authors · 2021-08-25
> > > **Thanks for the feedback**
> > >
> > > Thank you very much for the positive feedback. We highly appreciate that.

---

### Official Review · Reviewer_FR4z · 2021-07-20

**Rating:** 6
**Confidence:** 4

**Summary:**

This paper proposes a new approach for continual learning that actively forgets the old knowledge when learning the new tasks. The approach uses Bayesian continual learning with synaptic expansion-convergence for active forgetting. The paper evaluates the new method on several continual learning benchmarks, including CIFAR10 regression, visual classification, and Atari reinforcement tasks.

**Limitations And Societal Impact:**

No potential negative societal impact.

**Main Review:**

The synaptic expansion-convergence idea for active forgetting proposed in this paper seems novel for continual learning, although it is quite related to the power EP method by Minka. My main concern with the paper is its clarity, especially in the main technical part of the method. In particular:
1. The description of the proposed method in Section 3.2 is not very clear. For example, where is \theta_e in Eq (10)? How is \theta_e constructed (e.g. by expanding the network architecture or other methods)? How are the weights merged to obtain the original architecture? I find that the illustration in Figure 2 is not understandable. More details should be added here.
2. The paper should give more details on how to derive Eq (9).
3. Since the new prior has been changed by Eq (6), how does that affect the computation of the Fisher Information matrix in Eq (10)? More details should be added here, especially when there are more than 2 tasks.
4. Additionally, more details should be given on how the method can be applied with MAS, SI, and RWALK.
5. The CIFAR10 regression task in Section 4.1 is not realistic. Why would we want to convert a classification task into such a regression task? It may be better to consider a real regression task. Furthermore, the paper reported average accuracy for this regression task but did not give any explanation on how this average accuracy is computed.

===

The authors' rebuttal has adequately addressed my concerns. So I increase my score to 6.

**Time Spent Reviewing:**

3

---

> ### Author Response · Authors · 2021-08-08
> **Author Response**
>
> Thank you for your valuable comments. Below, we provide point-to-point response to the comments, especially to the clarity of the main technical part.
>
> ***Q1: More details about Sec. 3.2:***
>
> A1: Thanks. $\theta_e$ is the parameter of a separate network, which is optimized by $L_B(\theta_e)$ to obtain the optimal parameters $\theta_e^*$ for the new task (please refer to lines 148-149) and then compute its Fisher Information matrix $F_e$ (detailed in Appendix C.1 and C.3). After obtaining $\theta_e^*$ and $F_e$, we can optimize the main network parameter $\theta$ by Eq (10). After training the current task, $\theta_e^*$ (i.e., the separate network) and $F_e$ are deleted, so they will not cause any additional storage cost. When learning the next task, another set of $\theta_e$ is created to repeat the above procedures. We will add a pseudo code of AFEC and make the details clearer in the final version.
>
> ***Q2: More details on how to derive Eq (9):***
>
> A2: Thanks. Please refer to Appendix C.3 Eq (28) for the details of deriving Eq (9). We will make it clearer in the final version.
>
> ***Q3: How does the new prior affect the computation of the Fisher Information matrix in Eq (10)?***
>
> A3: Please see lines 133-137 and Appendix C.2, where we prove that the new prior defined in Eq (6) is still Gaussian if $p(\theta|D_A)$ and $p(\theta)$ are both Gaussian. Therefore, the new prior can be propagated to more incremental tasks, and does not affect the computation of the Fisher Information matrix in Eq (10), which is further detailed in Appendix C.1 lines 563-570. We will make it clearer in the final version.
>
> ***Q4: More details on how the method can be applied with MAS, SI, and RWALK:***
>
> A4: Thanks. Please see Appendix F for the details of how our method is adapted to MAS, SI, and RWALK. We will make it clearer in the final version.
>
> ***Q5: Better to consider a real regression task:***
>
> A5: Thanks. It's not difficult to find a realistic regression task. However, our major motivation to design the CIFAR10 regression task, which is a continual learning scenario with mutual negative transfers between all the tasks, is to more clearly validate our idea about negative transfer in continual learning (please see lines 184-185, 190-193). Specifically, we can explicitly show how negative transfer affects continual learning, and how our method effectively addresses this challenging issue. We will clarify it in the final version.
>
> ***Q6: How to compute the average accuracy of regression tasks:***
>
> A6: We average the accuracy of all the tasks ever seen (detailed in Appendix E Eq (29)), and then average for 5 runs (line 188 and the caption of Table 1). We will make it clearer in the final version.

---

> > ### Author Response · Authors · 2021-08-28
> > **Look forward to further feedback**
> >
> > Dear Reviewer FR4z,
> >
> > We thank you again for the valuable comments. We're very much looking forward to hearing your further feedback on the response. As all the other reviewers found the response satisfying, we hope you might have the similar feeling and view this as sufficient reason to further raise your score.
> >
> > Best,
> >
> > Authors

---

> > > ### Comment · Reviewer_FR4z · 2021-09-01
> > > **Thanks for the rebuttal**
> > >
> > > Thanks for the rebuttal and the additional experiments. They have adequately addressed my concerns. I will increase my score to 6. Please revise your paper accordingly.

---

> > > > ### Author Response · Authors · 2021-09-01
> > > > **Thank you for the feedback**
> > > >
> > > > Thank you so much for your positive feedback! We will incorporate the additional information and improve the paper in the final version.

---

### Author Response · Authors · 2021-08-08
**Our paper contributes to both continual learning and neuroscience.**

We thank all reviewers for their valuable comments. We are pleased that the reviewers found our paper ''novel'' (reviewer FR4z, WUG1), ''well-written'' (reviewer kQj5, we6u) and ''technically sound'' (reviewer we6u), with ''thoroughly conducted experiments'' (reviewer kQj5). In the individual rebuttals below, we provide point-to-point response to the comments.

In addition to the contribution of our paper to continual learning, which has been extensively discussed by all reviewers, we would like to highlight our contribution to neuroscience, which is illustrated in lines 44-64, 77-79, 152-155, Conclusion and Appendix A. In this paper, inspired by biological active forgetting, we propose to actively forget the old knowledge in the framework of Bayesian continual learning to mitigate negative transfer. Then we can derive an algorithm that is formally consistent with the synaptic expansion and synaptic convergence, which is the underlying mechanism of active forgetting in biological neural networks. The efficacy of our method provides a potential theoretical explanation for an important yet under addressed neuroscience problem: how the underlying mechanism of biological active forgetting achieves its function of forgetting the past and continually learning conflicting experiences [1-3]. Also, to the best of our knowledge, we are the first to model biological active forgetting and its underlying mechanism, which can potentially bridge the two fields of biological and artificial neural networks.

Thus, in addition to continual learning, our contribution to neuroscience will be a good fit to the audience of NeurIPS.



[1] Dong, T., He, J., et al. (2016). Inability to activate Rac1-dependent forgetting contributes to behavioral inflexibility in mutants of multiple autism-risk genes. PNAS, 113(27), 7644-7649.

[2] Davis, R. L., \& Zhong, Y. (2017). The biology of forgetting—a perspective. Neuron, 95(3), 490-503.

[3] Zhang, X., Li, Q., et al. (2018). Active protection: Learning-activated Raf/MAPK activity protects labile memory from Rac1-independent forgetting. Neuron, 98(1), 142-155.

---

### Author Response · Authors · 2021-08-21
**Author Response**

Dear reviewers:

Thanks a lot for your efforts in reviewing this paper. We tried our best to address all mentioned concerns/problems. Are there unclear explanations here? We could further clarify them.

Best,

Authors

---

### Author Response · Authors · 2021-08-26
**Compare with P&C in Pure Continual Learning Setting**

Dear AC and all Reviewers:

Although P&C did not release their code, we have tried our best to reproduce it in our setting of pure continual learning through adapting the implementation of Progressive Networks and online-EWC in the officially-released code of HAT [1] and AGS-CL [2]. We hope you might view this as sufficient reason to further raise your score.

Specifically, to make the comparison as fair as possible, the implementation details are the same as the baselines in our paper, such as architecture, number of epochs, learning rate, batch size, etc (detailed in Appendix D). We make an extensive grid search of the hyperparameter of P&C and report the best performance. Here we compare AFEC, EWC and P&C in visual classification benchmarks of CIFAR100-SC, CIFAR100 and CIFAR10/100 used in our paper. All the experiments are averaged over 5 runs with different random seeds and task orders.

#######################################################################################

Table 1. Averaged Accuracy (\%) of all the tasks learned so far. $A_t$ refers to the averaged accuracy of all the classes learned in t tasks.

|Method| CIFAR100-SC $A_{10}$| CIFAR100-SC $A_{20}$ | CIFAR100 $A_{10}$ | CIFAR100 $A_{20}$ |CIFAR10/100 $A_{2}$ | CIFAR10/100 $A_{2+20}$ |
| :----: | :----: | :----: | :----: | :----: | :----: | :----: |
|EWC              |52.25|51.74|68.72|69.18|85.07|77.75|
|P&C               |53.48|52.88|70.10|70.21|86.72|78.29|
|AFEC (ours) |56.28|55.24|72.36|72.29|86.87|81.25|

#######################################################################################

Table 2. Averaged forward transfer (FWT, \%) and backward transfer (BWT, \%).

|Method| CIFAR100-SC FWT| CIFAR100 FWT | CIFAR10/100 FWT| CIFAR100-SC BWT|CIFAR100 BWT|CIFAR10/100 BWT|
| :----: | :----: | :----: | :----: | :----: | :----: | :----: |
|EWC              |-7.39|-3.97|+1.36|-4.60|-5.79|-2.07|
|P&C               |-6.04|-1.96|+1.84|-4.65|-5.73|-2.21|
|AFEC (ours) |-3.37|-0.23|+4.47|-4.01|-5.75|-1.26|

#######################################################################################

As shown in Table 1, P&C outperforms EWC but substantially underperforms AFEC on the three visual classification benchmarks. We analyze the forward transfer (FWT) and backward transfer (BWT) in Table 2. FWT of P&C is better than EWC but lower than AFEC, while BWT of P&C and EWC are competing but slightly lower than AFEC. That is to say, without increasing catastrophic forgetting, AFEC achieves a much larger improvement on the learning of new tasks than P&C.

P&C penalizes the kl-divergence between the predictions of the knowledge base and the active column to *implicitly* forget the old knowledge in the knowledge base, and *non-selectively* acquire new knowledge from the active column.
In contrast, our active forgetting in AFEC is achieved by the weight-merging loss $\frac{\lambda_e}{2}\sum_i F_{e,i} (\theta_i - \theta_{e,i}^*)^2$ in Eq (10), where the forgetting factor is integrated into $\lambda_e$ to *explicitly* control the forgetting, while the Fisher Information $F_{e}$ enables the *selectivity* of acquiring knowledge from the expanded parameters. Therefore, AFEC more effectively improves the learning of new tasks and thus outperforms P&C.

We are still running P&C in other visual classification benchmarks, such as CUB-200 and ImageNet-100, and will extensively compare AFEC and P&C in the final version.

[1] Serra, J., Suris, D., et al. (2018). Overcoming catastrophic forgetting with hard attention to the task. ICML 2018.

[2] Jung, S., Ahn, H., et al. (2020). Continual learning with node-importance based adaptive group sparse regularization. NeurIPS 2020.

---

> ### Author Response · Authors · 2021-08-27
> **Additional Results on Large-Scale Images**
>
> We further compare P&C, EWC and AFEC in the other three visual classification benchmarks used in our paper, such as CUB-200 with ImageNet pretraining (refers to as CUB-200 w/ PT), CUB-200 w/o PT and ImageNet-100, which are all on large-scale images. As shown in Table 3 below, the results are consistent with the one of three variants of CIFAR100 in Table 1 above. We will add it in the final version.
>
> #######################################################################################################
>
> Table 3. Averaged Accuracy (\%) of all the learned classes so far on large-scale images. $A_t$ refers to the averaged accuracy of all the classes learned in t tasks.
>
> |Method| CUB-200 w/ PT $A_{5}$| CUB-200 w/ PT $A_{10}$ | CUB-200 w/o PT $A_{5}$ | CUB-200 w/o PT $A_{10}$ |ImageNet-100 $A_{5}$ | ImageNet-100 $A_{10}$ |
> | :----: | :----: | :----: | :----: | :----: | :----: | :----: |
> |EWC              |81.37|80.92|32.90|42.29|76.12|73.82|
> |P&C               |81.42|81.74|33.88|42.79|76.44|74.38|
> |AFEC (ours) |83.65|82.04|34.36|43.05|77.64|75.46|
>
> #######################################################################################################

---

### Decision · Program_Chairs · 2021-09-27

**Decision:**

Accept (Poster)

**Comment:**

This work touches on an important problem in Continual Learning that has not been as thoroughly investigated as catastrophic forgetting, namely graceful forgetting (or an active and controlled mode of forgetting) that would allow forward transfer.
While the need of graceful forgetting has been discussed in the past, there are not many methods that manage to do so in a successful way, and this work proposes a practical way of doing just that.

Given the reviews, and the active participation of the authors in the discussion (including new experiments) I believe the current manuscript is of interest to the community and matches the expectations from the conference.